# Multiple Descent: Design Your Own Generalization Curve

## Abstract

This paper explores the generalization loss of linear regression in variably parameterized families of models, both under-parameterized and over-parameterized. We show that the generalization curve can have an arbitrary number of peaks, and moreover, locations of those peaks can be explicitly controlled. Our results highlight the fact that both classical U-shaped generalization curve and the recently observed double descent curve are not intrinsic properties of the model family. Instead, their emergence is due to the interaction between the properties of the data and the inductive biases of learning algorithms.

## 1 Introduction

The main goal of machine learning methods is to provide an accurate out-of-sample prediction, known as generalization. For a fixed family of models, a common way to select a model from this family is through empirical risk minimization, i.e., algorithmically selecting models that minimize the risk on the training dataset. Given a variably parameterized family of models, the statistical learning theory aims to identify the dependence between model complexity and model performance. The empirical risk usually decreases monotonically as the model complexity increases, and achieves its minimum when the model is rich enough to interpolate the training data, resulting in zero (or near-zero) training error. In contrast, the behaviour of the test error as a function of model complexity is far more complicated. Indeed, in this paper we show how to construct a model family for which the generalization curve can be fully controlled (away from the interpolation threshold) in both under-parameterized and over-parameterized regimes. Classical statistical learning theory supports a U-shaped curve of generalization versus model complexity (Geman et al., 1992; Hastie et al., 2009). Under such a framework, the best model is found at the bottom of the U-shaped curve, which corresponds to appropriately balancing under-fitting and over-fitting the training data. From the view of the bias-variance trade-off, a higher model complexity increases the variance while decreasing the bias. A good choice of model complexity achieves a relatively low bias while still keeping the variance under control. On the other hand, a model that interpolates the training data is deemed to over-fit and tends to worsen the generalization performance due to the soaring variance.

Although classical statistical theory suggests a pattern of behavior for the generalization curve up to the interpolation threshold, it does not describe what happens beyond the interpolation threshold, commonly referred to as the over-parameterized regime. This is the exact regime where many modern machine learning models, especially deep neural networks, achieved remarkable success. Indeed, neural networks generalize well even when the models are so complex that they have the potential to interpolate all the training data points (Zhang et al., 2017; Belkin et al., 2018b; Ghorbani et al., 2019; Hastie et al., 2019).

Modern practitioners commonly deploy deep neural networks with hundreds of millions or even billions of parameters. It has become widely accepted that large models achieve performance superior to small models that may be suggested by the classical U-shaped generalization curve (Bengio et al., 2003; Krizhevsky et al., 2012; Szegedy et al., 2015; He et al., 2016; Huang et al., 2019). This indicates that the test error decreases again once model complexity grows beyond the interpolation threshold, resulting in the so called double-descent phenomenon described in (Belkin et al., 2018a), which has been broadly supported by empirical evidence (Neyshabur et al., 2015; Neal et al., 2018; Geiger et al., 2019; 2020) and confirmed empirically on modern neural architectures by Nakkiran et al. (2019). On the theoretical side, this phenomenon has been recently addressed by several works

on various model settings. In particular, Belkin et al. (2019a) proved the existence of double-descent phenomenon for linear regression with random feature selection and analyzed the random Fourier feature model (Rahimi & Recht, 2008). Mei & Montanari (2019) also studied the Fourier model and computed the asymptotic test error which captures the double-descent phenomenon. Bartlett et al. (2020); Tsigler & Bartlett (2020) analyzed and gave explicit conditions for "benign overfitting" in linear and ridge regression, respectively. In a recent work, Caron & Chretien (2020) provided a finite sample analysis of the nonlinear function estimation and showed that the parameter learned through empirical risk minimization converges to the true parameter with high probability as the model complexity tends to infinity, implying the existence of double descent.

Among all the aforementioned efforts, one particularly interesting question is whether one can observe more than two descents in the generalization curve. In a recent work, d'Ascoli et al. (2020) empirically showed a sample-wise triple-descent phenomenon under the random Fourier feature model. Similar triple-descent was also observed for linear regression (Nakkiran et al., 2020). More rigorously, Liang et al. (2020) presented an upper bound on the risk of the minimum-norm interpolation versus the data dimension in Reproducing Kernel Hilbert Spaces (RKHS), which exhibits multiple descent. However, a multiple-descent upper bound without a properly matching lower bound does not imply the existence of a multiple-descent generalization curve. In this work, we study the multiple descent phenomenon by addressing the following questions:

- Can the existence of a multiple descent generalization curve be rigorously proven?
- Can an arbitrary number of descents occur?
- Can the generalization curve and the locations of descents be designed?

In this paper, we show that the answer to all three of these questions is yes. Further related work is presented in Appendix A.

**Our Contribution.** We consider the linear regression model and analyze how the risk changes as the dimension of the data grows. In the linear regression setting, the data dimension is equal to the dimension of the parameter space, which reflects the model complexity. We rigorously show that the multiple descent generalization curve exists under this setting. To our best knowledge, this is the first work proving a multiple descent phenomenon for any learning model.

Our analysis considers both the underparameterized and overparameterized regimes. In the overparameterized regime, we show that one can control where a descent or an ascent occurs in the generalization curve. This is realized through our algorithmic construction of a feature-revealing process. To be more specific, we assume that the data is in $\mathbb{R}^D$, where $D$ can be arbitrarily large or even essentially infinite. We view each dimension of the data as a feature. We consider a linear regression problem restricted on the first $d$ features, where $d < D$. New features are revealed by increasing the dimension of the data. We then show that by specifying the distribution of the newly revealed feature to be either a standard Gaussian or a Gaussian mixture, one can determine where an ascent or a descent occurs. In order to create an ascent when a new feature is revealed, it is sufficient that the feature follows a Gaussian mixture distribution. In order to have a descent, it is sufficient that the new feature follows a standard Gaussian distribution. Therefore, in the overparameterized regime, we can fully control the occurrence of a descent and an ascent. As a comparison, in the underparameterized regime, the generalization loss always increases regardless of the feature distribution. We also consider a dimension-normalized version of the generalization loss, under which we show that the generalization curve exhibits multiple descent in the underparameterized regime. Generally speaking, we show that we are able to design the generalization curve.

On the one hand, we show theoretically that the generalization curve is malleable and can be constructed in an arbitrary fashion. On the other hand, we rarely observe complex generalization curves in practice, besides carefully curated constructions. Putting these facts together, we arrive at the conclusion that realistic generalization curves arise from specific interactions between properties of typical data and the inductive biases of algorithms. We should highlight that the nature of these interactions is far from being understood and should be an area of further investigations.

## 2 PRELIMINARIES AND PROBLEM FORMULATION

**Notation.** For $x \in \mathbb{R}^D$ and $d \leq D$, we let $x[1:d] \in \mathbb{R}^d$ denote a $d$-dimensional vector with $x[1:d]_i = x_i$ for all $1 \leq i \leq d$. For a matrix $A \in \mathbb{R}^{n \times d}$, we denote its Moore-Penrose pseudoinverse by

$A^+ \in \mathbb{R}^{d \times n}$. We use the big O notation $\mathcal{O}$ and write variables in the subscript of $\mathcal{O}$ if the implicit constant depends on them. For example, $\mathcal{O}_{n,d,\sigma}(1)$ is a constant that only depends on $n$, $d$, and $\sigma$. If $f(\sigma)$ and $g(\sigma)$ are functions of $\sigma$, write $f(\sigma) \sim g(\sigma)$ if $\lim \frac{f(\sigma)}{g(\sigma)} = 1$. It will be given in the context how we take the limit.

**Distributions.** Let $\mathcal{N}(\mu, \sigma^2)$ ($\mu, \sigma \in \mathbb{R}$) and $\mathcal{N}(\mu, \Sigma)$ ($\mu \in \mathbb{R}^n$, $\Sigma \in \mathbb{R}^{n \times n}$) denote the univariate and multivariate Gaussian distributions, respectively, where $\mu \in \mathbb{R}^n$ and $\Sigma \in \mathbb{R}^{n \times n}$ is a positive semi-definite matrix. We define a family of *trimodal* Gaussian mixture distributions as follows

$$\mathcal{N}_{\sigma,\mu}^{\mathrm{mix}} \triangleq \frac{1}{3}\mathcal{N}(0, \sigma^2) + \frac{1}{3}\mathcal{N}(-\mu, \sigma^2) + \frac{1}{3}\mathcal{N}(\mu, \sigma^2).$$

For an illustration, please see Fig. 1.

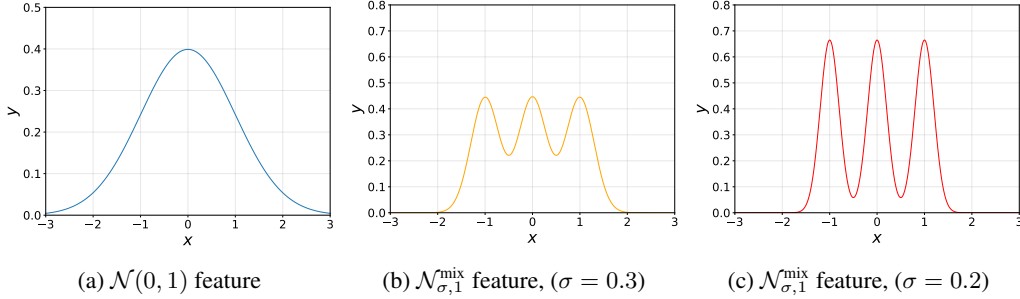

(a) $\mathcal{N}(0,1)$ feature     (b) $\mathcal{N}_{\sigma,1}^{\mathrm{mix}}$ feature, $(\sigma = 0.3)$     (c) $\mathcal{N}_{\sigma,1}^{\mathrm{mix}}$ feature, $(\sigma = 0.2)$

Figure 1: Density functions of the $\mathcal{N}(0,1)$ and $\mathcal{N}_{\sigma,1}^{\mathrm{mix}}$ feature. A new entry is independently sampled from the 1-dimensional distribution being either a standard Gaussian or trimodal Gaussian mixture. Smaller $\sigma$ leads to higher concentration around each modes.

Let $\chi^2(k, \lambda)$ denote the noncentral chi-squared distribution with $k$ degrees of freedom and the non-centrality parameter $\lambda$. For example, if $X_i \sim \mathcal{N}(\mu_i, 1)$ (for $i = 1, 2, \ldots, k$) are independent Gaussian random variables, we have $\sum_{i=1}^{k} X_i^2 \sim \chi^2(k, \lambda)$, where $\lambda = \sum_{i=1}^{k} \mu_i^2$. We also denote by $\chi^2(k)$ the (central) chi-squared distribution with $k$ degrees and the $F$-distribution by $F(d_1, d_2)$ where $d_1$ and $d_2$ are the degrees of freedom.

**Problem Setup.** Let $x_1, \ldots, x_n \in \mathbb{R}^D$ be column vectors that represent the training data of size $n$ and let $x_{\mathrm{test}} \in \mathbb{R}^D$ be a column vector that represents the test data. We assume that they are all independently drawn from a distribution

$$x_1, \ldots, x_n, x_{\mathrm{test}} \stackrel{iid}{\sim} \mathcal{D}.$$

Let us consider a linear regression problem on the first $d$ features, where $d \leq D$ for some arbitrary large $D$. Here, $d$ can be viewed as the number of features revealed. The *design* matrix $A$ equals $[x_1[1:d], \ldots, x_n[1:d]]^\top \in \mathbb{R}^{n \times d}$. The true linear model is $\beta^* \in \mathbb{R}^d$. The noise $\varepsilon \in \mathbb{R}^n$ follows the multivariate standard Gaussian distribution $\mathcal{N}(0, \eta^2 I_n)$. Let $x = x_{\mathrm{test}}[1:d]$ denote the first $d$ features of the test data.

For the underparameterized regime where $d < n$, the least square solution on the training data is $A^+(A\beta^* + \varepsilon)$. For the overparameterized regime where $d > n$, $A^+(A\beta^* + \varepsilon)$ is the minimum-norm solution. In both regimes we consider the solution $\hat{\beta} \triangleq A^+(A\beta^* + \varepsilon)$. The excess generalization loss on the test data is then given by

$$\begin{aligned}
L_d &\triangleq \mathbb{E}\left[\left(y - x^\top \hat{\beta}\right)^2 - \left(y - x^\top \beta^*\right)^2\right] \\
&= \mathbb{E}\left[\left(x^\top(\hat{\beta} - \beta^*)\right)^2\right] \\
&= \mathbb{E}\left[\left(x^\top\left((A^+A - I)\beta^* + A^+\varepsilon\right)\right)^2\right] \\
&= \mathbb{E}\left[(x^\top(A^+A - I)\beta^*)^2\right] + \mathbb{E}\left[(x^\top A^+\varepsilon)^2\right] \\
&= \mathbb{E}\left[(x^\top(A^+A - I)\beta^*)^2\right] + \eta^2\mathbb{E}\left\|(A^\top)^+x\right\|^2,
\end{aligned}$$

where $y = x^\top \beta^* + \varepsilon_{\text{test}}$ and $\varepsilon_{\text{test}} \sim \mathcal{N}(0, \eta^2)$. We call the term $\mathbb{E}\left[(x^\top (A^+ A - I)\beta^*)^2\right]$ the *bias* and call the term $\eta^2 \mathbb{E} \left\| (A^\top)^+ x \right\|^2$ the *variance*.

In this paper, we assume $\beta^* = 0$ and the noise level $\eta = 1$. In this settings, we get

$$L_d = \mathbb{E}\|(A^\top)^+ x\|^2.$$

**Remark 1.** In the underparametrized regime, if $\mathcal{D}$ is a continous distribution (our construction presented later satisfies this condition), the matrix $A$ has independent column almost surely. In this case, we have $A^+ A = I$ and therefore the bias $\mathbb{E}\left[(x^\top (A^+ A - I)\beta^*)^2\right]$ vanishes irrespective of the true linear model $\beta^*$. In other words, in the underparametrized regime, $L_d$ equals $\eta^2 \mathbb{E}\|(A^\top)^+ x\|^2$ for all $\beta^*$.

We would like to study the change in the loss caused by the growth in the number of features revealed. Note that the product $(A^+)^\top x$ sums over $d$ dimensions. Once we reveal a new feature, which is equivalent to adding a new row $b^\top$ to $A^\top$ and a new component $y$ to $x$, the product $\begin{bmatrix} A^\top \\ b^\top \end{bmatrix}^+ \begin{bmatrix} x \\ y \end{bmatrix}$ sums over $d + 1$ dimensions. As a result, to compare quantities of different dimensions, we need to normalize the generalization loss by the dimension. We define the dimension-normalized generalization loss $L'_d$ as follows

$$L'_d \triangleq \mathbb{E} \left\| \frac{1}{d}(A^\top)^+ x \right\|^2 = \frac{1}{d^2} L_d.$$

**Local Maximum and Multiple Descent.** We say that a local maximum occurs at a dimension $d \geq 1$ if $L'_{d-1} < L'_d$ and $L'_d > L'_{d+1}$. Intuitively, a local maximum occurs if there is an increasing stage of the generalization loss, followed by a decreasing stage, as the dimension $d$ grows. Additionally, we define $L'_0 \triangleq -\infty$. If the generalization loss exhibits a single descent, based on our definition, a unique local maximum occurs at $d = 1$. For a double-descent generalization curve, a local maximum occurs at two different dimensions. In general, if we observe a local maximum at $K$ different dimensions we call it a $K$-descent.

## 3  UNDERPARAMETERIZED REGIME

First, we present our main theorem for the underparametrized regime below, whose proof is deferred to the end of Section 3. It states that the un-normalized generalization loss $L_d$ is always nondecreasing as $d$ grows. Moreover, it is possible to have an arbitrarily large ascent, i.e., $L_{d+1} - L_d > C$ for any $C > 0$.

**Theorem 1** (Proof in Appendix B.1). *If $d < n$, we have $L_{d+1} \geq L_d$ irrespective of the data distribution. Moreover, for any $C > 0$, there exists a distribution $\mathcal{D}$ such that $L_{d+1} - L_d > C$.*

For the dimension-normalized generalization loss $L'_d$, there can be both ascents and descents. And it is possible to specify where the local peaks in the generalization curve occur.

**Theorem 2** (Underparameterized regime). *Let $D + 2 < \sqrt{2n}$. For any $1 < d_1 < d_2 < \cdots < d_K < D$ where $d_{j+1} - d_j \geq 2$, there exists a distribution $\mathcal{D}$ such that a local maximum of the $L'_d$ curve occurs at $d_j$.*

Note that the assumption $d_{j+1} - d_j \geq 2$ is necessary because two local maxima may not be adjacent. We present an example in Fig. 2.

**Remark 2** ($\mathcal{D}$ can be a product distribution). As will be clear later in the proof of Theorem 2, the distribution $\mathcal{D}$ can be made as simple as a product distribution $\mathcal{D} = \mathcal{D}_1 \times \cdots \times \mathcal{D}_D$ such that $x_{i,j} \overset{iid}{\sim} \mathcal{D}_j$ for all $1 \leq i \leq n$, where $\mathcal{D}_j$ is either sampled from $\mathcal{N}(0, 1)$ or a Gaussian mixture $\mathcal{N}_{\sigma_j}^{\text{mix}}$ for some $\sigma_j > 0$. As a consequence, by permuting the order of $\mathcal{D}_i$'s in the product distribution, we can change the order of revealing the features.

**Remark 3** (Kernel regression on Gaussian data). In light of Remark 2, $\mathcal{D}$ can be chosen to be a product distribution that consists of only $\mathcal{N}(0, 1)$ and $\mathcal{N}_{\sigma_j}^{\text{mix}}$. Note that one can simulate $\mathcal{N}_{\sigma,1}^{\text{mix}}$

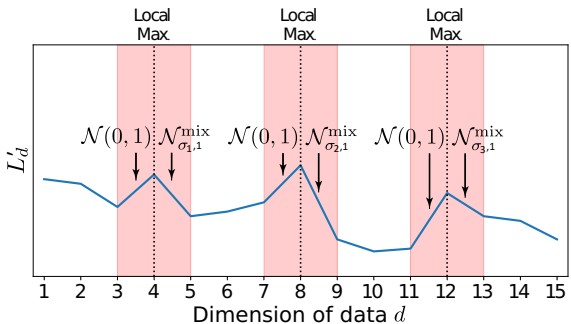

Figure 2: Illustration of multiple descent for the dimension-normalized generalization loss $L'_d$ as a function of the dimension $d$. A local maximum occurs at $d$ if $L'_{d-1} < L'_d > L'_{d+1}$. The triplet $L'_{d-1}$, $L'_d$, $L'_{d+1}$ then form an ascent/descent, which is marked by the shaded area. Local maxima are marked by the dotted lines. Adding a new feature with a Gaussian mixture distribution increases the loss, while adding one with a univariate Gaussian distribution decreases the loss. Therefore, a Gaussian mixture feature followed by a Gaussian feature creates one ascent/descent.

with $\mathcal{N}(0,1)$ through the inverse transform sampling. To see this, let $F_{\mathcal{N}(0,1)}$ and $F_{\mathcal{N}^{\text{mix}}_{\sigma,1}}$ be the cdf of $\mathcal{N}(0,1)$ and $\mathcal{N}^{\text{mix}}_{\sigma,1}$, respectively. If $X \sim \mathcal{N}(0,1)$, we have $F_{\mathcal{N}(0,1)}(X) \sim \text{Unif}((0,1))$ and therefore $\varphi_\sigma(X) \triangleq F^{-1}_{\mathcal{N}^{\text{mix}}_{\sigma,1}}(F_{\mathcal{N}(0,1)}(X)) \sim \mathcal{N}^{\text{mix}}_{\sigma,1}$. In fact, we can use a multivariate Gaussian $\mathcal{D}' = \mathcal{N}(0, I_{D \times D})$ and a sequence of non-linear kernels $k^{[1:d]}(x,y) \triangleq \langle \phi^{[1:d]}(x), \phi^{[1:d]}(y) \rangle$, where the feature map is $\phi^{[1:d]}(x) \triangleq [\phi_1(x_1), \phi_2(x_2), \ldots, \phi_d(x_d)]^\top \in \mathbb{R}^d$. Here is a simple rule for defining $\phi_j$: If $\mathcal{D}_j = \mathcal{N}(0,1)$, we set $\phi_j$ to the identity function. If $\mathcal{D}_j = \mathcal{N}^{\text{mix}}_{\sigma_j}$, we set $\phi_j$ to $\varphi_{\sigma_j}$. Thus, the problem becomes a kernel regression problem on the standard Gaussian data.

Getting back Theorem 2, let us discuss how we will construct such a distribution $\mathcal{D}$ inductively. We fix $d$. Again, denote the first $d$ features of $x_{\text{test}}$ by $x \triangleq x_{\text{test}}[1 : d]$. Let us consider adding an additional component to the training data $x_1[1 : d], \ldots, x_n[1 : d]$ and test data $x$ so that we increment the dimension $d$ by 1. Let $b_i \in \mathbb{R}$ denote the additional component that we add to the vector $x_i$ (so that the new vector is $[x_i[1 : d]^\top, b_i]^\top$. Similarly, let $y \in \mathbb{R}$ denote the additional component that we add to the vector $x$. We form the column vector $b = [b_1, \ldots, b_n]^\top \in \mathbb{R}^n$ that collects all additional components that we add to the training data.

We consider the change in (dimension-normalized) generalization loss as follows

$$L_{d+1} - L_d = \mathbb{E}\left[\left\|\begin{bmatrix} A^\top \\ b^\top \end{bmatrix}^+ \begin{bmatrix} x \\ y \end{bmatrix}\right\|^2 - \left\|(A^+)^\top x\right\|^2\right],$$

$$L'_{d+1} - L'_d = \mathbb{E}\left[\left\|\frac{1}{d+1}\begin{bmatrix} A^\top \\ b^\top \end{bmatrix}^+ \begin{bmatrix} x \\ y \end{bmatrix}\right\|^2 - \left\|\frac{1}{d}(A^+)^\top x\right\|^2\right].$$

(1)

Note that the components $b_1, \ldots, b_n, y$ are i.i.d. Lemma 3 relates the pseudo-inverse of $[A, b]^\top$ to that of $A^\top$.

**Lemma 3** (Proof in Appendix B.2). *Let $A \in \mathbb{R}^{n \times d}$ and $0 \neq b \in \mathbb{R}^{n \times 1}$, where $n \geq d + 1$. Additionally, let $P = AA^+$ and $Q = bb^+ = \frac{bb^\top}{\|b\|^2}$, and define $z \triangleq \frac{b^\top(I-P)b}{\|b\|^2}$. If $z \neq 0$ and the columnwise partitioned matrix $[A, b]$ has linearly independent columns, we have*

$$\begin{bmatrix} A^\top \\ b^\top \end{bmatrix}^+ = \left[\left(I - \frac{bb^\top}{\|b\|^2}\right)\left(I + \frac{AA^+bb^\top}{\|b\|^2 - b^\top AA^+b}\right)(A^+)^\top, \frac{(I-AA^+)b}{\|b\|^2 - b^\top AA^+b}\right]$$

$$= \left[(I-Q)(I + \frac{PQ}{1-\text{tr}(PQ)})(A^+)^\top, \frac{(I-P)b}{b^\top(I-P)b}\right]$$

$$= \left[(I-Q)(I + \frac{PQ}{z})(A^+)^\top, \frac{(I-P)b}{b^\top(I-P)b}\right].$$

In our construction of $\mathcal{D}$, the components $\mathcal{D}_j$ are all continuous distributions. The matrix $I - P$ is an orthogonal projection matrix and therefore $\operatorname{rank}(I - P) = n - d$. As a result, it holds almost surely that $b \neq 0$, $z \neq 0$, and $[A, b]$ has linearly independent columns. Thus the assumptions of Lemma 3 are satisfied almost surely. In the sequel, we assume that these assumptions are always fulfilled.

**Lemma 4** (Proof in Appendix B.3). *Assume $d$, $n > d + 2$ and $P$ are fixed, where $P \in \mathbb{R}^{n \times n}$ is an orthogonal projection matrix whose rank is $d$. Define $z \triangleq \frac{b^\top (I - P) b}{\|b\|^2}$, where $b = [b_1, \dots, b_n]^\top \in \mathbb{R}^n$. If $y$, $b_1, \cdots, b_n \overset{iid}{\sim} \mathcal{N}_{\sigma,1}^{\mathrm{mix}}$, we have $\mathbb{E}[1/z] = \mathcal{O}_{n,d,\sigma}(1)$ and $\mathbb{E}[y^2 / b^\top (I - P) b] = \mathcal{O}_{n,d,\sigma}(1)$.*

Theorem 5 provides an upper bound for the following quantity

$$\mathbb{E}_{b,y}\left[\left\|\frac{1}{d+1}\begin{bmatrix} A^\top \\ b^\top \end{bmatrix}^+ \begin{bmatrix} x \\ y \end{bmatrix}\right\|^2 - \left\|\frac{1}{d}(A^+)^\top x\right\|^2 \middle| A, x\right]$$

if $b_1, \dots, b_n, y$ are i.i.d. according to $\mathcal{N}(0, 1)$ or $\mathcal{N}_{\sigma,1}^{\mathrm{mix}}$. This quantiry is similar to the difference between the dimension-normalized generalization loss $L'_{d+1} - L'_d$ but with expectation only over $b$ and $y$.

**Theorem 5** (Proof in Appendix B.4). *Conditioned on $A$ and $x$, the following statements hold:*

(a) *If $d + 2 < \sqrt{2n}$ and $b_1, \dots, b_n, y \overset{iid}{\sim} \mathcal{N}(0, 1)$, we have*

$$\mathbb{E}_{b,y}\left[\left\|\frac{1}{d+1}\begin{bmatrix} A^\top \\ b^\top \end{bmatrix}^+ \begin{bmatrix} x \\ y \end{bmatrix}\right\|^2 - \left\|\frac{1}{d}(A^+)^\top x\right\|^2\right] < \frac{d - \|(A^+)^\top x\|^2 (2n - (d+2)^2)}{d(d+1)^2(n - d - 2)} . \quad (2)$$

(b) *If $d + 2 < n$ and $b_1, \dots, b_n, y \overset{iid}{\sim} \mathcal{N}_{\sigma,1}^{\mathrm{mix}}$, we have*

$$\mathbb{E}_{b,y}\left\|\frac{1}{d+1}\begin{bmatrix} A^\top \\ b^\top \end{bmatrix}^+ \begin{bmatrix} x \\ y \end{bmatrix}\right\|^2 \leq \left\|\frac{1}{d}(A^+)^\top x\right\|^2 + \mathcal{O}_{n,d,\sigma}(1),$$

*where $\mathcal{O}_{n,d,\sigma}(1)$ is a universal constant that only depends on $n$, $d$, and $\sigma$.*

**Corollary 6.** *Assume $d + 2 < \sqrt{2n}$. If either $b_1, \dots, b_n, y \overset{iid}{\sim} \mathcal{N}(0, 1)$ or $b_1, \dots, b_n, y \overset{iid}{\sim} \mathcal{N}_{\sigma,1}^{\mathrm{mix}}$, and by taking expectation over all random variables, we have*

$$\mathbb{E}\left\|\frac{1}{d+1}\begin{bmatrix} A^\top \\ b^\top \end{bmatrix}^+ \begin{bmatrix} x \\ y \end{bmatrix}\right\|^2 = \mathcal{O}_{n,d,\sigma}\left(\mathbb{E}\left\|\frac{1}{d}(A^+)^\top x\right\|^2\right).$$

We will use Theorem 5 in two different ways. The first way is presented in Corollary 6. We would like to show inductively (on $d$) that $L'_d$ is finite for every $d$. Provided that we are able to guarantee finite $L'_1$, Corollary 6 implies that $L'_d$ is finite for every $d$ if the components are always sampled from $\mathcal{N}(0, 1)$ or $\mathcal{N}_{\sigma,1}^{\mathrm{mix}}$.

Alternatively, we can use Theorem 5 to create a descent, i.e., make $L'_{d+1} < L'_d$. In light of (2), to make the left-hand side negative, we need

$$d - \|(A^+)^\top x\|^2 (2n - (d+2)^2) < 0,$$

which is equivalent to

$$\left\|\frac{1}{d}(A^+)^\top x\right\|^2 > \frac{1}{d(2n - (d+2))^2} .$$

One we take expectation over $A$ and $x$, we need the above equation to hold in expectation in order to create a descent, i.e.,

$$L'_d > \frac{1}{d(2n - (d+2))^2} .$$

Provided that $L'_d$ can be made sufficiently large, letting $L'_D$ satisfy the above inequality and then adding an additional $\mathcal{N}(0, 1)$ entry will lead to $L'_{d+1} < L'_d$. Making a large $L'_d$, in turn, can be achieved by adding an entry sampled from $\mathcal{N}_{\sigma,1}^{\mathrm{mix}}$ when the data dimension increases from $d - 1$ to $d$ in the previous step. Indeed, Theorem 7 shows that adding a $\mathcal{N}_{\sigma,1}^{\mathrm{mix}}$ feature can increase the loss by arbitrary amount.

**Theorem 7** (Proof in Appendix B.5). *For any $C > 0$ and $\mathbb{E}\left\|(A^+)^\top x\right\|^2 < +\infty$, there exists a $\sigma > 0$ such that if $b_1, \ldots, b_n, y \overset{iid}{\sim} \mathcal{N}_{\sigma,1}^{\text{mix}}$, we have*

$$\mathbb{E}\left[\left\|\begin{bmatrix} A^\top \\ b^\top \end{bmatrix}^+ \begin{bmatrix} x \\ y \end{bmatrix}\right\|^2 - \left\|(A^+)^\top x\right\|^2\right] > C,$$

$$\mathbb{E}\left[\left\|\frac{1}{d+1}\begin{bmatrix} A^\top \\ b^\top \end{bmatrix}^+ \begin{bmatrix} x \\ y \end{bmatrix}\right\|^2 - \left\|\frac{1}{d}(A^+)^\top x\right\|^2\right] > C.$$

We are now ready to prove Theorem 2.

*Proof of Theorem 2.* We construct $\mathcal{D}$ inductively. Let $\mathcal{D}_1 = \mathcal{N}(0,1)$. When $d = 1$, we have

$$A = [x_1[1:d], \ldots, x_n[1:d]]^\top = [x_{1,1}, \ldots, x_{n,1}]^\top \in \mathbb{R}^n,$$

which is a column vector. Therefore, $A^+ = \frac{A^\top}{\|A\|^2}$. As a result, we get

$$L_1' = \mathbb{E}\left\|(A^+)^\top x\right\|^2 = \mathbb{E}\frac{|x|^2}{\|A\|^2} = \frac{1}{n-2},$$

where $x \sim \mathcal{N}(0,1)$ and $\|A\|^2 \sim \chi^2(n)$.

Since we will set $\mathcal{D}_j$ (for $j \geq 2$) to either $\mathcal{N}(0,1)$ or $\mathcal{N}_{\sigma,1}^{\text{mix}}$, by Corollary 6, we have $L_{j+1}' = \mathcal{O}_{n,j,\sigma_{j+1}}(L_j')$. By induction, we obtain that $L_j'$ is finite for all $1 \leq j \leq D$.

We define $d_0 \triangleq 0$. Assume that we have determined distributions $\mathcal{D}_1, \ldots, \mathcal{D}_{d_j+1}$, where $0 \leq j < K$. We set $\mathcal{D}_{d_j+2}, \ldots, \mathcal{D}_{d_{j+1}-1}$ to $\mathcal{N}(0,1)$. For $\mathcal{D}_{d_{j+1}}$, by Theorem 7, we pick $\sigma_{d_{j+1}}$ such that if $\mathcal{D}_{d_{j+1}} = \mathcal{N}_{\sigma_{d_{j+1}}}^{\text{mix}}$, we have

$$L_{d_{j+1}}' > \max\left\{L_{d_{j+1}-1}', \frac{1}{d_{j+1}(2n - (d_{j+1}+2)^2)}\right\}. \tag{3}$$

Next, we set $\mathcal{D}_{d_{j+1}+1} = \mathcal{N}(0,1)$. Taking the expectation of (2) in Theorem 5 over all random variables, we have

$$L_{d_{j+1}+1}' - L_{d_{j+1}}' \leq \frac{d_{j+1} - d_{j+1}^2 L_{d_{j+1}}'(2n - (d_{j+1}+2)^2)}{d_{j+1}(d_{j+1}+1)^2(n - d_{j+1} - 2)} < 0,$$

where the last inequality is due to (3). So far we have constructed a local maximum at $d_{j+1}$. By induction, we conclude that a local maximum occurs at every $d_j$. $\qquad\qquad\square$

**Remark 4.** From Remark 2 and the proof of Theorem 2 it is clear that $\mathcal{D} = \mathcal{D}_1 \times \cdots \times \mathcal{D}_D$ is a product distribution. The construction in the proof also shows that the generalization curve is actually determined by the specific choice of the $\mathcal{D}_i$'s. Note that permuting the order of $\mathcal{D}_i$'s is equivalent to changing the order by which the features are being revealed (i.e., permuting the entries of the data $x_i$'s). Therefore, given the same data points $x_1, \cdots, x_n \in \mathbb{R}^D$, we can create many different generalization curves simply by changing the order of the feature-revealing process.

## 4   OVERPARAMETERIZED REGIME

In this section, we study the multiple decent phenomenon in the overparameterized regime. Note that as stated in Section 2, we consider the minimum-norm solution here. As stated in the following theorem, we require $d \geq n+8$, which means $d$ starts at roughly the same order as $n$. In other words, the result covers almost the entire spectrum of the overparameterized regime.

**Theorem 8** (Overparameterized regime). *Let $n < D - 9$. Given any sequence $\Delta_{n+8}, \Delta_{n+9}, \ldots, \Delta_{D-1}$ where $\Delta_d \in \{\uparrow, \downarrow\}$, there exists a distribution $\mathcal{D}$ such that for every $n + 8 \leq d \leq D - 1$, we have*

$$L_{d+1}\begin{cases} > L_d, & \text{if } \Delta_d = \uparrow \\ < L_d, & \text{if } \Delta_d = \downarrow \end{cases}, \quad L_{d+1}'\begin{cases} > L_d', & \text{if } \Delta_d = \uparrow \\ < L_d', & \text{if } \Delta_d = \downarrow \end{cases}.$$

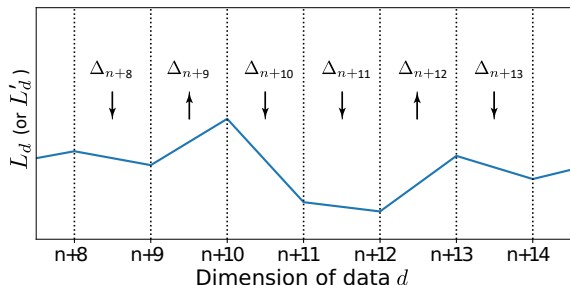

Figure 3: Illustration of the multiple descent phenomenon for the generalization loss $L_d$ (or the dimension-normalized generalization loss $L'_d$) versus the dimension of data $d$ in the overparameterized regime starting from $d = n + 8$. One can fully control the generalization curve to increase or decrease as specified by the sequence $\Delta = \{\downarrow, \uparrow, \downarrow, \downarrow, \uparrow, \downarrow, \dots\}$. Adding a new feature with Gaussian mixture distribution increases the loss, while adding one with Gaussian distribution decreases the loss.

In Theorem 8, the sequence $\Delta_{n+8}$, $\Delta_{n+9}$, $\cdots$, $\Delta_{D-1}$ is just used to specify the increasing/decreasing behavior of the $L'_d$ sequence for $d > n + 8$. Compared to Theorem 2 for the underparameterized regime, where one is able to fully control the ascents but only partially control the descents, Theorem 8 indicates that one is able to fully control both ascents and descents in the overparameterized regime by placing an ascent/descent wherever one desires. Fig. 3 illustrates an example.

Lemma 9 gives the pseudo-inverse of $A$ when $d > n$.

**Lemma 9** (Proof in Appendix C.1). *Let $A \in \mathbb{R}^{n \times d}$ and $b \in \mathbb{R}^{n \times 1}$, where $n \leq d$. Assume that matrix $A$ and the columnwise partitioned matrix $B \triangleq [A, b]$ have linearly independent rows. Let $G \triangleq (AA^\top)^{-1} \in \mathbb{R}^{n \times n}$ and $u \triangleq \frac{b^\top G}{1 + b^\top G b} \in \mathbb{R}^{1 \times n}$. We have*

$$\begin{bmatrix} A^\top \\ b^\top \end{bmatrix}^+ = \left[ (I - bu)^\top (A^+)^\top, u^\top \right] .$$

Lemma 10 establishes finite expectation for several random variables. These finite expectation results are necessary for Theorem 11 and Theorem 12 to hold. Technically, they are the dominating random variables needed in Lebesgue's dominated convergence theorem. Lemma 10 indicates that to guarantee these finite expectations, it suffices to set the first $n + 8$ distributions to the standard normal distribution and then set $\mathcal{D}_{n+8}, \dots, \mathcal{D}_D$ to either a Gaussian or a Gaussian mixture distribution. In fact, in Theorem 11 and Theorem 12, we always add a Gaussian distribution or a Gaussian mixture.

**Lemma 10** (Proof in Appendix C.2). *Let $\mathcal{D} = \mathcal{D}_1 \times \cdots \times \mathcal{D}_D$ be a product distribution where*

(a) *$\mathcal{D}_d = \mathcal{N}(0, 1)$ if $d = 1, \dots, n + 8$; and*

(b) *$\mathcal{D}_d$ is either $\mathcal{N}(0, \sigma_d^2)$ or $\mathcal{N}_{\sigma_d, \mu_d}^{\mathrm{mix}}$ for $d > n + 8$.*

*Let $\mathcal{D}_{[1:d]}$ denote $\mathcal{D}_1 \times \cdots \times \mathcal{D}_d$. Assume that every row of $A \in \mathbb{R}^{n \times d}$ and $x \in \mathbb{R}^{d \times 1}$ are i.i.d. and follow $\mathcal{D}_{[1:d]}$. For any $d$ such that $n + 8 \leq d \leq D$, all of the followings hold:*

$$\begin{aligned}
\mathbb{E}[\|(A^+)^\top x\|^2] &< +\infty, \\
\mathbb{E}[\lambda_{\max}^2((AA^\top)^{-1})] &< +\infty, \\
\mathbb{E}[\lambda_{\max}((AA^\top)^{-1})\|(A^+)^\top x\|^2] &< +\infty, \\
\mathbb{E}[\lambda_{\max}^2((AA^\top)^{-1})\|(A^+)^\top x\|^2] &< +\infty.
\end{aligned} \tag{4}$$

Theorem 11 shows that in order to have $L_{d+1} < L_d$ and $L'_{d+1} < L'_d$, it suffices to add a Gaussian feature.

**Theorem 11** (Appendix C.3). *If $\mathbb{E}[\|(A^\top A)^+ x\|^2] > 0$ and all equations in* (4) *hold, there exists $\sigma > 0$ such that if $y, b_1, \ldots, b_n \overset{iid}{\sim} \mathcal{N}(0, \sigma^2)$, we have*

$$L_{d+1} - L_d = \mathbb{E}\left\|\begin{bmatrix} A^\top \\ b^\top \end{bmatrix}^+ \begin{bmatrix} x \\ y \end{bmatrix}\right\|^2 - \mathbb{E}\left\|(A^+)^\top x\right\|^2 < 0\,,$$

$$L'_{d+1} - L'_d = \mathbb{E}\left\|\frac{1}{d+1}\begin{bmatrix} A^\top \\ b^\top \end{bmatrix}^+ \begin{bmatrix} x \\ y \end{bmatrix}\right\|^2 - \mathbb{E}\left\|\frac{1}{d}(A^+)^\top x\right\|^2 < 0\,.$$

Theorem 12 shows that adding a Gaussian mixture feature can make $L_{d+1} > L_d$ and $L'_{d+1} > L'_d$.

**Theorem 12** (Proof in Appendix C.4). *Assume $\mathbb{E}\|(A^+)^\top x\|^2 < +\infty$. For any $C > 0$, there exist $\mu, \sigma > 0$ such that if $y, b_1, \ldots, b_n \overset{iid}{\sim} \mathcal{N}^{\text{mix}}_{\sigma,\mu}$, we have*

$$L_{d+1} - L_d = \mathbb{E}\left\|\begin{bmatrix} A^\top \\ b^\top \end{bmatrix}^+ \begin{bmatrix} x \\ y \end{bmatrix}\right\|^2 - \mathbb{E}\left\|(A^+)^\top x\right\|^2 > C\,,$$

$$L'_{d+1} - L'_d = \mathbb{E}\left\|\frac{1}{d+1}\begin{bmatrix} A^\top \\ b^\top \end{bmatrix}^+ \begin{bmatrix} x \\ y \end{bmatrix}\right\|^2 - \mathbb{E}\left\|\frac{1}{d}(A^+)^\top x\right\|^2 > C\,.$$

The proof of Theorem 8 immediately follows from Theorem 11 and Theorem 12.

*Proof of Theorem 8.* We construct the product distribution $\mathcal{D} = \prod_{d=1}^{D} \mathcal{D}_d$. We set $\mathcal{D}_d = \mathcal{N}(0, 1)$ for $d = 1, \ldots, n+8$. For $n + 8 < d \leq D$, $\mathcal{D}_d$ is either $\mathcal{N}(0, \sigma_d^2)$ or $\mathcal{N}^{\text{mix}}_{\sigma_d, \mu_d}$ depending on $\Delta_d$ being either $\downarrow$ or $\uparrow$.

First we show that for each step $d$, the assumption $\mathbb{E}[\|(A^\top A)^+ x\|^2] > 0$ of Theorem 11 is satisfied. If $\mathbb{E}[\|(A^\top A)^+ x\|^2] = 0$, we know that $(A^\top A)^+ x = 0$ almost surely. Since $\mathcal{D}$ is a continuous distribution, the matrix $A$ has full row rank almost surely. Therefore, $\text{rank}((A^\top A)^+) = \text{rank}(A^\top A) = n$ almost surely. Thus $\dim \ker(A^\top A)^+ = d - n \leq d - 1$ almost surely, which implies $x \notin \ker(A^\top A)^+$. In other words, $(A^\top A)^+ x \neq 0$ almost surely. We reach a contradiction. Moreover, by Lemma 10, the assumption $\mathbb{E}\|(A^+)^\top x\|^2 < +\infty$ of Theorem 12 is also satisfied.

If $\Delta_{d-1} = \downarrow$, by Theorem 11, there exists $\sigma_d > 0$ such that if $\mathcal{D}_d = \mathcal{N}(0, \sigma_d^2)$, then $L_d < L_{d-1}$ and $L'_d < L'_{d-1}$. Similarly if $\Delta_{d-1} = \uparrow$, by Theorem 12, there exists $\sigma_d$ and $\mu_d$ such that $\mathcal{D}_d = \mathcal{N}^{\text{mix}}_{\sigma_d, \mu_d}$ guarantees $L_d > L_{d-1}$ and $L'_d > L'_{d-1}$.

$\square$

## 5   CONCLUSION

Our work proves that the expected risk of linear regression can manifest multiple descents when the number of features increases and sample size is fixed. This is carried out through an algorithmic construction of a feature-revealing process where the newly revealed feature follows either a Gaussian distribution or a Gaussian mixture distribution. Notably, the construction also enables us to control local maxima in the underparameterized regime and control ascents/descents freely in the overparameterized regime. Overall, this allows us to design the generalization curve away from the interpolation threshold. We conjecture that the same multiple-descent generalization curve can occur in non-linear neural networks and we humbly suggest that entities with infinite computational powers investigate this phenomenon.

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

## A    FURTHER RELATED WORK

Our work is directly related to the recent line of research in the theoretical understanding of the double descent (Belkin et al., 2019a; Hastie et al., 2019; Xu & Hsu, 2019; Mei & Montanari, 2019) and the multiple descent phenomenon (Liang et al., 2020). Here we briefly discuss some other work that is closely related to this paper.

**Least Square Regression.**    In this paper we focus on the least square linear regression with no regularization. For the regularized least square regression, De Vito et al. (2005) proposed a selection procedure for the regularization parameter. Advani & Saxe (2017) analyzed the generalization of neural networks with mean squared error under the asymptotic regime where both the sample size and model complexity tend to infinity. Richards et al. (2020) proved for least square regression in the asymptotic regime that as the dimension-to-sample-size ratio $d/n$ grows, an additional peak can occur in both the variance and bias due to the covariance structure of the features. As a comparison, in this paper the sample size is fixed and the model complexity increases. Rudi & Rosasco (2017) studied kernel ridge regression and gave an upper bound on the number of the random features to reach certain risk level. Our result shows that there exists a natural setting where by manipulating the random features one can control the risk curve.

**Over-Parameterization and Interpolation.**    The double descent occurs when the model complexity reaches and increases beyond the interpolation threshold. Most previous works focused on proving an upper bound or optimal rate for the risk. Caponnetto & De Vito (2007) gave the optimal rate for least square ridge regression via careful selection of the regularization parameter. Belkin et al. (2019b) showed that the optimal rate for risk can be achieved by a model that interpolates the training data. In a series of work on kernel regression with regularization parameter tending to zero (a.k.a. kernel *ridgeless* regression), Rakhlin & Zhai (2019) showed that the risk is bounded away from zero when the data dimension is fixed with respect to the sample size. Liang & Rakhlin (2019) then considered the case when $d \asymp n$ and proved a risk upper bound that can be small given favorable data and kernel assumptions. Instead of giving a bound, our paper presents an exact computation of risk in the cases of underparameterized and overparameterized linear regression, and proves the existence of the multiple descent phenomenon. Wyner et al. (2017) analyzed AdaBoost and Random Forest from the perspective of interpolation. There has also been a line of work on wide neural networks (Arora et al., 2019a;b;c; Du et al., 2019; Allen-Zhu et al., 2019; Wei et al., 2019; Cao & Gu, 2019; Advani et al., 2020; Chen & Xu, 2020; Zou et al., 2020).

**Sample-wise Double Descent and Non-monotonicity.**    There has also been recent development beyond the model-complexity double-descent phenomenon. For example, regarding sample-wise non-monotonicity, Nakkiran et al. (2019) empirically observed the epoch-wise double-descent and sample-wise non-monotonicity for neural networks. Chen et al. (2020) and Min et al. (2020) identified and proved the sample-wise double descent under the adversarial training setting, and Javanmard et al. (2020) discovered double-descent under adversarially robust linear regression. Loog et al. (2019) showed that empirical risk minimization can lead to sample-wise non-monotonicity in the standard linear model setting under various loss functions including the absolute loss and the squared loss, which covers the range from classification to regression. We also refer the reader to their discussion of the earlier work on non-monotonicity of generalization curves. Dar et al. (2020) demonstrated the double descent curve of the generalization errors of subspace fitting problems.

## B    PROOFS FOR UNDERPARAMETRIZED REGIME

### B.1    PROOF OF THEOREM 1

*Proof.* We follow the notation convention in (1):

$$L_{d+1} - L_d = \mathbb{E}\left[\left\|\begin{bmatrix} A^\top \\ b^\top \end{bmatrix}^+ \begin{bmatrix} x \\ y \end{bmatrix}\right\|^2 - \left\|(A^\top)^+ x\right\|^2\right].$$

Recall $d < n$ and the matrix $B' \triangleq \begin{bmatrix} A^\top \\ b^\top \end{bmatrix}$ is of size $(d+1) \times n$. Both matrices $B'$ and $B \triangleq A^\top$ are fat matrices. As a result, if $x' \triangleq \begin{bmatrix} x \\ y \end{bmatrix}$, we have

$$\|B'^+ x'\|^2 = \min_{z:B'z=x'} \|z\|^2, \quad \|B^+ x\|^2 = \min_{z:Bz=x} \|z\|^2.$$

Since $\{z \mid B'z = x'\} \subseteq \{z \mid Bz = x\}$, we get $\|B'^+ x'\|^2 \geq \|B^+ x\|^2$. Therefore, we obtain $L_{d+1} \geq L_d$.

The second part of the theorem, which says that for any $C > 0$ there exists a distribution such that $L_{d+1} - L_d > C$, follows directly from Theorem 7. □

## B.2 PROOF OF LEMMA 3

*Proof.* By (Baksalary & Baksalary, 2007, Theorem 1), we have

$$\begin{bmatrix} A^\top \\ b^\top \end{bmatrix}^+ = \left[ (I - Q)A(A^\top(I-Q)A)^{-1}, \ \frac{(I-P)b}{b^\top(I-P)b)} \cdot \right]$$

Define $r \triangleq A^\top b \in \mathbb{R}^d$. Since $A$ has linearly independent columns, the Gram matrix $G = A^\top A$ is non-singular. The Sherman-Morrison formula gives

$$(A^\top(I-Q)A)^{-1} = \left( A^\top A - \frac{rr^\top}{\|b\|^2} \right)^{-1} = G^{-1} + \frac{G^{-1}rr^\top G^{-1}}{\|b\|^2 - r^\top G^{-1}r} = G^{-1} + \frac{G^{-1}rb^\top(A^+)^\top}{\|b\|^2 - r^\top G^{-1}r},$$

where we use the facts $r = A^\top b$ and $AG^{-1} = (A^+)^\top$ in the last equality. Therefore, we deduce

$$\begin{aligned} A(A^\top(I-Q)A)^{-1} &= AG^{-1} + \frac{AG^{-1}rb^\top(A^+)^\top}{\|b\|^2 - r^\top G^{-1}r} \\ &= (A^+)^\top + \frac{AG^{-1}A^\top bb^\top(A^+)^\top}{\|b\|^2 - r^\top G^{-1}r} \\ &= \left( I + \frac{AA^+bb^\top}{\|b\|^2 - r^\top G^{-1}r} \right)(A^+)^\top \\ &= \left( I + \frac{PQ}{1 - \frac{r^\top G^{-1}r}{\|b\|^2}} \right)(A^+)^\top. \end{aligned}$$

Observe that

$$1 - \frac{r^\top G^{-1}r}{\|b\|^2} = 1 - \frac{b^\top A(A^\top A)^{-1}A^\top b}{\|b\|^2} = 1 - \frac{b^\top Pb}{\|b\|^2} = z.$$

Therefore, we obtain the desired expression. □

## B.3 PROOF OF LEMMA 4

Lemma 13 shows that a noncentral $\chi^2$ distribution first-order stochastically dominates a central $\chi^2$ distribution of the same degree of freedom. It will be needed in the proof of Lemma 4.

**Lemma 13.** *Assume that random variables $X \sim \chi^2(k, \lambda)$ and $Y \sim \chi^2(k)$, where $\lambda > 0$. For any $c > 0$, we have*

$$\mathbb{P}(X \geq c) > \mathbb{P}(Y \geq c).$$

*In other words, the random variable $X$ (first-order) stochastically dominates $Y$.*

*Proof.* Let $Y_1, X_2, \ldots, X_k \overset{iid}{\sim} \mathcal{N}(0,1)$ and $X_1 \sim \mathcal{N}(\sqrt{\lambda}, 1)$ and all these random variables are jointly independent. Then $X' \triangleq \sum_{i=1}^k X_i^2 \sim \chi^2(k, \lambda)$ and $Y' \triangleq Y_1^2 + \sum_{i=2}^k X_i^2 \sim \chi^2(k)$.

It suffices to show that $\mathbb{P}(X' \geq c) > \mathbb{P}(Y' \geq c)$, or equivalently, $\mathbb{P}(|\mathcal{N}(\mu, 1)| \geq c) > \mathbb{P}(|\mathcal{N}(0, 1)| \geq c)$ for all $c > 0$ and $\mu \triangleq \sqrt{\lambda} > 0$. Denote $F_c(t) = \mathbb{P}(|\mathcal{N}(\mu, 1)| \geq c)$ and we have

$$F_c(\mu) = 1 - \frac{1}{\sqrt{2\pi}} \int_{-c}^{c} \exp\left(-\frac{(x-\mu)^2}{2}\right) dx = 1 - \frac{1}{\sqrt{2\pi}} \int_{-c-\mu}^{c-\mu} \exp\left(-\frac{x^2}{2}\right) dx,$$

and thus

$$\frac{dF_c(\mu)}{d\mu} = \frac{1}{\sqrt{2\pi}} \left[\exp\left(-\frac{(c-\mu)^2}{2}\right) - \exp\left(-\frac{(c+\mu)^2}{2}\right)\right] > 0.$$

This shows $\mathbb{P}(|\mathcal{N}(\mu, 1)| \geq c) > \mathbb{P}(|\mathcal{N}(0, 1)| \geq c)$ and we are done.

$\square$

*Proof of Lemma 4.* Since $b_i \overset{iid}{\sim} \mathcal{N}_{\sigma,1}^{\mathrm{mix}}$, we can rewrite $b = u + w$ where $w \sim \mathcal{N}(0, \sigma^2 I_n)$ and the entries of $u$ satisfy $u_i \overset{iid}{\sim} \mathrm{Unif}(\{-1, 0, 1\})$. Furthermore, $u$ and $w$ are independent. Note that for any fixed $n$ and $d$, the support of $u$ is finite and its cardinality only depends on $n$. Therefore, we only need to show that conditioning on $u$, the expectation over $w$ is $\mathcal{O}_{n,d,\sigma}(1)$. In other words, for any fixed $u$, we want to show $\mathbb{E}_w[1/z \mid u] = \mathcal{O}_{n,d,\sigma}(1)$ and $\mathbb{E}_w\left[\frac{y^2}{b^\top(I-P)b} \Big| u\right] = \mathcal{O}_{n,d,\sigma}(1)$.

Note that since $y^2/\sigma^2$ is first-order stochastically dominated by $\chi^2(1, 1)$, we have

$$\mathbb{E}[y^2 \mid u] = \mathbb{E}[y^2] \leq \sigma^2 \mathbb{E}[\chi^2(1, 1)] = 2\sigma^2.$$

Therefore, it remains to show $\mathbb{E}_w[1/z \mid u] = \mathcal{O}_{n,d,\sigma}(1)$ and $\mathbb{E}_w\left[\frac{1}{b^\top(I-P)b} \Big| u\right] = \mathcal{O}_{n,d,\sigma}(1)$.

Note that

$$\frac{1}{z} = \frac{b^\top I b}{b^\top(I-P)b} = 1 + \frac{(u+w)^\top P(u+w)}{(u+w)^\top(I-P)(u+w)}.$$

Since $P$ is an orthogonal projection, there exists an orthogonal transformation $O$ depending only on $P$ such that

$$(u+w)^\top P(u+w) = [O(u+w)]^\top D_d[O(u+w)]$$

where $D_d = \mathrm{diag}([1, \ldots, 1, 0 \ldots, 0])$ with $d$ diagonal entries equal to 1 and the others equal to 0. We denote $\tilde{u} = O(u)$, which is fixed (as $u$ and $O$ are fixed), and $\tilde{w} = O(w) \sim \mathcal{N}(0, \sigma^2 I_n)$. It follows that

$$\frac{1}{z} = 1 + \frac{(\tilde{u}+\tilde{w})^\top D_d(\tilde{u}+\tilde{w})}{(\tilde{u}+\tilde{w})^\top(I-D_d)(\tilde{u}+\tilde{w})} = 1 + \frac{\sum_{i=1}^{d}(\tilde{u}_i+\tilde{w}_i)^2}{\sum_{i=d+1}^{n}(\tilde{u}_i+\tilde{w}_i)^2} = 1 + \frac{\sum_{i=1}^{d}(\tilde{u}_i+\tilde{w}_i)^2/\sigma^2}{\sum_{i=d+1}^{n}(\tilde{u}_i+\tilde{w}_i)^2/\sigma^2}.$$

Observe that

$$\sum_{i=1}^{d}(\tilde{u}_i+\tilde{w}_i)^2/\sigma^2 \sim \chi^2\left(d, \sqrt{\sum_{i=1}^{d}\tilde{u}_i^2}\right)$$

$$\sum_{i=d+1}^{n}(\tilde{u}_i+\tilde{w}_i)^2/\sigma^2 \sim \chi^2\left(n-d, \sqrt{\sum_{i=d+1}^{n}\tilde{u}_i^2}\right),$$

and that these two quantities are independent. It follows that

$$\mathbb{E}\left[\sum_{i=1}^{d}(\tilde{u}_i+\tilde{w}_i)^2/\sigma^2 \Big| u\right] = d + \sqrt{\sum_{i=1}^{d}\tilde{u}_i^2}.$$

By Lemma 13, the denominator $\sum_{i=d+1}^{n}(\tilde{u}_i+\tilde{w}_i)^2/\sigma^2$ first-order stochastically dominates $\chi^2(n-d)$. Therefore, we have

$$\mathbb{E}\left[\frac{1}{\sum_{i=d+1}^{n}(\tilde{u}_i+\tilde{w}_i)^2/\sigma^2} \Big| u\right] \leq \mathbb{E}\left[\frac{1}{\chi^2(n-d)}\right] = \frac{1}{n-d-2}.$$

Putting the numerator and denominator together yields

$$\mathbb{E}\left[\frac{1}{z}\bigg|u\right] \le 1 + \frac{d + \sqrt{\sum_{i=1}^{d}\tilde{u}_i^2}}{n - d - 2} \le 1 + \frac{d + \sqrt{d}}{n - d - 2} = \mathcal{O}_{n,d,\sigma}(1)\,.$$

Similarly, we have

$$
\begin{aligned}
\mathbb{E}\left[\frac{1}{b^\top(I-P)b}\bigg|u\right] &= \mathbb{E}\left[\frac{1}{[O(u+w)]^\top(I-D_d)[O(u+w)]}\bigg|u\right] \\
&= \mathbb{E}\left[\frac{1/\sigma^2}{\sum_{i=d+1}^{n}(\tilde{u}_i + \tilde{w}_i)^2/\sigma^2}\bigg|u\right] \\
&\le \frac{1}{\sigma^2}\mathbb{E}\left[\frac{1}{\chi^2(n-d)}\right] \\
&= \frac{1}{\sigma^2}\cdot\frac{1}{n-d-2} \\
&= \mathcal{O}_{n,d,\sigma}(1)\,.
\end{aligned}
$$

$\square$

### B.4   PROOF OF THEOREM 5

*Proof.* First, we rewrite the expression as follows

$$
\begin{aligned}
&\left\|\frac{1}{d+1}\begin{bmatrix}A^\top\\b^\top\end{bmatrix}^+\begin{bmatrix}x\\y\end{bmatrix}\right\|^2 - \left\|\frac{1}{d}(A^+)^\top x\right\|^2 \\
&= \frac{1}{(d+1)^2}\left\|(I-Q)(I+PQ/z)(A^+)^\top x + \frac{(I-P)b}{b^\top(I-P)b}y\right\|^2 - \frac{1}{d^2}\|(A^+)^\top x\|^2\,,
\end{aligned}
\tag{5}
$$

where $P, Q, z$ are defined in Lemma 3. Since $y$ has mean 0 and is independent of other random variables, so that the cross term vanishes under expectation over $b$ and $y$:

$$\mathbb{E}_{b,y}\left[\left\langle (I-Q)(I+PQ/z)(A^+)^\top x, \frac{(I-P)b}{b^\top(I-P)b}y\right\rangle\right] = 0\,,$$

where $\langle\cdot,\cdot\rangle$ denotes the inner product. Therefore taking the expectation of (5) over $b$ and $y$ yields

$$
\mathbb{E}_{b,y}\left[\left\|\frac{1}{d+1}\begin{bmatrix}A^\top\\b^\top\end{bmatrix}^+\begin{bmatrix}x\\y\end{bmatrix}\right\|^2 - \left\|\frac{1}{d}(A^+)^\top x\right\|^2\right]
\tag{6}
$$

$$
= \mathbb{E}_{b,y}\left[\frac{1}{(d+1)^2}\|(I-Q)(I+PQ/z)(A^+)^\top x\|^2 - \frac{1}{d^2}\|(A^+)^\top x\|^2 + \frac{1}{(d+1)^2}\left\|\frac{(I-P)b}{b^\top(I-P)b}y\right\|^2\right]
\tag{7}
$$

$$
= \frac{1}{(d+1)^2}\mathbb{E}_{b,y}\left[\|(I-Q)(I+PQ/z)(A^+)^\top x\|^2 - (1+\tfrac{1}{d})^2\|(A^+)^\top x\|^2 + \left\|\frac{(I-P)b}{b^\top(I-P)b}y\right\|^2\right]\,.
\tag{8}
$$

We simplify the third term. Recall that $I - P = I - AA^+$ is an orthogonal projection matrix and thus idempotent

$$\left\|\frac{(I-P)b}{b^\top(I-P)b}y\right\|^2 = \frac{y^2}{(b^\top(I-P)b)^2}\|(I-P)b\|^2 = \frac{y^2}{b^\top(I-P)b}\,. \tag{9}$$

Thus we have

$$
\mathbb{E}_{b,y}\left[\left\|\frac{1}{d+1}\begin{bmatrix}A^\top\\b^\top\end{bmatrix}^+\begin{bmatrix}x\\y\end{bmatrix}\right\|^2 - \left\|\frac{1}{d}(A^+)^\top x\right\|^2\right] \tag{10}
$$

$$
= \frac{1}{(d+1)^2}\mathbb{E}_{b,y}\left[\|(I-Q)(I+PQ/z)(A^+)^\top x\|^2 - (1+\frac{1}{d})^2\|(A^+)^\top x\|^2 + \frac{y^2}{b^\top(I-P)b}\right]. \tag{11}
$$

We consider the first and second terms. We write $v = (A^+)^\top x$ and define $z = \frac{b^\top(I-P)b}{\|b\|^2}$. The sum of the first and second terms equals

$$
\|(I-Q)(I+PQ/z)v\|^2 - (1+\frac{1}{d})^2\|v\|^2 = -v^\top(M+\delta I)v\,, \tag{12}
$$

where $\delta = \frac{2}{d} + \frac{1}{d^2}$ and

$$
M \triangleq Q - \frac{PQ+QP}{z} + \left(\frac{2}{z}-\frac{1}{z^2}\right)QPQ + \frac{QPQPQ}{z^2}\,.
$$

The rank of $M$ is at most 2. To see this, we re-write $M$ in the following way

$$
M = \left[Q\left(-\frac{P}{z}+\left(\frac{2}{z}-\frac{1}{z^2}\right)PQ+\frac{PQPQ}{z^2}\right)\right] + \left[-\frac{PQ}{z}\right] \triangleq M_1 + M_2\,.
$$

Notice that $\operatorname{rank}(M_1) \leq \operatorname{rank}(Q)$, $\operatorname{rank}(M_2) \leq \operatorname{rank}(Q)$, and $\operatorname{rank}(Q) = 1$. It follows that $\operatorname{rank}(M) \leq \operatorname{rank}(M_1) + \operatorname{rank}(M_2) = 2$. The matrix $M$ has at least $n-2$ zero eigenvalues. We claim that $M$ has two non-zero eigenvalues and they are $1 - 1/z < 0$ and 1.

Since

$$
\operatorname{rank}(PQ) \leq \operatorname{rank}(Q) = 1
$$

and

$$
\operatorname{tr}(PQ) = \frac{b^\top Pb}{\|b\|^2} = 1 - z,
$$

thus $PQ$ has a unique non-zero eigenvalue $1 - z$. Let $u \neq 0$ denote the corresponding eigenvector such that $PQu = (1-z)u$. Since $u \in \operatorname{im}P$ and $P$ is a projection, we have $Pu = u$. Therefore we can verify that

$$
Mu = (1-\frac{1}{z})u\,.
$$

To show that the other non-zero eigenvalue of $M$ is 1, we compute the trace of $M$

$$
\operatorname{tr}(M) = \operatorname{tr}(Q) - \frac{2\operatorname{tr}(PQ)}{z} + \left(\frac{2}{z}-\frac{1}{z^2}\right)\operatorname{tr}(PQ) + \frac{\operatorname{tr}((PQ)^2)}{z^2} = 2 - \frac{1}{z}\,,
$$

where we use the fact that $\operatorname{tr}(Q) = 1$, $\operatorname{tr}(PQ) = 1 - z$,

$$
\operatorname{tr}((PQ)^2) = \operatorname{tr}\left(\frac{Pbb^\top Pbb^\top}{\|b\|^4}\right) = \operatorname{tr}\left(\frac{(b^\top Pb)(b^\top Pb)}{\|b\|^4}\right) = (1-z)^2\,.
$$

We have shown that $M$ has eigenvalue $1 - 1/z$ and $M$ has at most two non-zero eigenvalues. Therefore, the other non-zero eigenvalue is $\operatorname{tr}(M) - (1 - 1/z) = 1$.

We are now in a position to upper bound (12) as follows:

$$
-v^\top(M+\delta I)v \leq -(1 - 1/z + \delta)\|v\|^2 < -(1 - 1/z + 2/d)\|v\|^2\,.
$$

Putting all three terms of the change in the dimension-normalized generalization loss yields

$$
\mathbb{E}_{b,y}\left[\left\|\frac{1}{d+1}\begin{bmatrix}A^\top\\b^\top\end{bmatrix}^+\begin{bmatrix}x\\y\end{bmatrix}\right\|^2 - \left\|\frac{1}{d}(A^+)^\top x\right\|^2\right]
$$

$$
\leq \frac{1}{(d+1)^2}\mathbb{E}_{b,y}\left[-(1 - 1/z + 2/d)\|v\|^2 + \frac{y^2}{b^\top(I-P)b}\right]\,.
$$

For $b_1, \ldots, b_n, y \overset{iid}{\sim} \mathcal{N}(0,1)$, we have $\mathbb{E}[y^2] = 1$. Moreover, $b^\top (I - P)b$ follows $\chi^2(n - d)$ a distribution. Thus $\frac{1}{b^\top (I-P)b}$ follows an inverse-chi-squared distribution with mean $\frac{1}{n-d-2}$. Therefore the expectation $\mathbb{E}[\frac{y^2}{b^\top (I-P)b}] = \frac{1}{n-d-2}$. Notice that $1/z$ follows a $1 + \frac{d}{n-d} F(d, n-d)$ distribution and thus $\mathbb{E}[1/z] = 1 + \frac{d}{n-d-2}$. As a result, we obtain

$$\mathbb{E}_{b,y} \left[ \left\| \frac{1}{d+1} \begin{bmatrix} A^\top \\ b^\top \end{bmatrix}^+ \begin{bmatrix} x \\ y \end{bmatrix} \right\|^2 - \left\| \frac{1}{d}(A^+)^\top x \right\|^2 \right]$$

$$\leq \frac{1}{(d+1)^2} \left[ \left( \frac{d}{n-d-2} - \frac{2}{d} \right) \|v\|^2 + \frac{1}{n-d-2} \right]$$

$$< \frac{d - \|v\|^2 (2n - (d+2)^2)}{d(d+1)^2(n-d-2)} .$$

For $b_1, \ldots, b_n, y \overset{iid}{\sim} \mathcal{N}_{\sigma,1}^{\mathrm{mix}}$, Lemma 4 implies that

$$\mathbb{E}_{b,y}[1/z] < \mathcal{O}_{n,d,\sigma}(1),$$

and

$$\mathbb{E}_{b,y} \left[ \frac{y^2}{b^\top (I - P)b} \right] < \mathcal{O}_{n,d,\sigma}(1).$$

Therefore, we conclude that

$$\mathbb{E}_{b,y} \left\| \frac{1}{d+1} \begin{bmatrix} A^\top \\ b^\top \end{bmatrix}^+ \begin{bmatrix} x \\ y \end{bmatrix} \right\|^2 \leq \left\| \frac{1}{d}(A^+)^\top x \right\|^2 + \mathcal{O}_{n,d,\sigma}(1).$$

$\square$

## B.5 Proof of Theorem 7

*Proof.* We start from (11). Taking expectation over all random variables gives

$$\mathbb{E} \left[ \left\| \frac{1}{d+1} \begin{bmatrix} A^\top \\ b^\top \end{bmatrix}^+ \begin{bmatrix} x \\ y \end{bmatrix} \right\|^2 - \left\| \frac{1}{d}(A^+)^\top x \right\|^2 \right]$$

$$= \frac{1}{(d+1)^2} \mathbb{E} \left[ \|(I-Q)(I+PQ/z)(A^+)^\top x\|^2 - (1+\frac{1}{d})^2 \|(A^+)^\top x\|^2 + \frac{y^2}{b^\top (I-P)b} \right]$$

$$\geq \frac{1}{(d+1)^2} \left( -(1+\frac{1}{d})^2 \mathbb{E}\|(A^+)^\top x\|^2 + \mathbb{E} \left[ \frac{y^2}{\sum_{i=1}^n b_i^2} \right] \right).$$

Our strategy is to choose $\sigma$ so that $\mathbb{E}\left[ \frac{y^2}{\sum_{i=1}^n b_i^2} \right]$ is sufficiently large. This is indeed possible as we immediately show. Define independent random variables $u \sim \mathrm{Unif}(\{-1,0,1\})$ and $w \sim \mathcal{N}(0,\sigma^2)$. Since $y$ has the same distribution as $u + w$, we have

$$\mathbb{E}[y^2] = \mathbb{E}[(u+w)^2] = \mathbb{E}[u^2] + \mathbb{E}[w^2] \geq \frac{2}{3}.$$

On the other hand,

$$\mathbb{E} \left[ \frac{1}{\sum_{i=1}^n b_i^2} \right] \geq \mathbb{P}(\max_i |b_i| \leq \sigma) \, \mathbb{E} \left[ \frac{1}{\sum_{i=1}^n b_i^2} \Big| \max_i |b_i| \leq \sigma \right]$$

$$= [\mathbb{P}(|b_1| \leq \sigma)]^n \, \mathbb{E} \left[ \frac{1}{\sum_{i=1}^n b_i^2} \Big| \max_i |b_i| \leq \sigma \right]$$

$$\geq \left[ \frac{1}{3\sqrt{2\pi\sigma^2}} \int_{-\sigma}^{\sigma} \exp\left( -\frac{t^2}{2\sigma^2} \right) dt \right]^n \frac{1}{n\sigma^2}$$

$$\geq \frac{1}{5^n n\sigma^2}.$$

Together we have

$$\mathbb{E}\left[\frac{y^2}{\sum_{i=1}^n b_i^2}\right] \geq \frac{1}{5^{n+1}n\sigma^2}\,.$$

Since

$$\mathbb{E}\left[\left\|\begin{bmatrix} A^\top \\ b^\top \end{bmatrix}^+ \begin{bmatrix} x \\ y \end{bmatrix}\right\|^2 - \|(A^+)^\top x\|^2\right] \geq d^2 \mathbb{E}\left[\left\|\frac{1}{d+1}\begin{bmatrix} A^\top \\ b^\top \end{bmatrix}^+ \begin{bmatrix} x \\ y \end{bmatrix}\right\|^2 - \left\|\frac{1}{d}(A^+)^\top x\right\|^2\right],$$

we have

$$\lim_{\sigma\to 0^+} \mathbb{E}\left[\left\|\begin{bmatrix} A^\top \\ b^\top \end{bmatrix}^+ \begin{bmatrix} x \\ y \end{bmatrix}\right\|^2 - \|(A^+)^\top x\|^2\right]$$

$$= \lim_{\sigma\to 0^+} \mathbb{E}\left[\left\|\frac{1}{d+1}\begin{bmatrix} A^\top \\ b^\top \end{bmatrix}^+ \begin{bmatrix} x \\ y \end{bmatrix}\right\|^2 - \left\|\frac{1}{d}(A^+)^\top x\right\|^2\right] = +\infty\,,$$

which completes the proof. $\qquad\square$

## C  PROOFS FOR OVERPARAMETRIZED REGIME

### C.1  PROOF OF LEMMA 9

*Proof.* Since $A$ and $B$ have full row rank, $(AA^\top)^{-1}$ and $(BB^\top)^{-1}$ exist. Therefore we have

$$B^+ = B^\top(BB^\top)^{-1}.$$

The Sherman-Morrison formula gives

$$(BB^\top)^{-1} = (AA^\top + bb^\top)^{-1} = G - \frac{Gbb^\top G}{1 + b^\top Gb} = G - Gbu = G(I - bu)\,.$$

Hence, we deduce

$$B^+ = [A, b]^\top G(I - bu) = \begin{bmatrix} A^\top G(I - bu) \\ b^\top G(I - bu) \end{bmatrix} = \begin{bmatrix} A^+(I - bu) \\ b^\top G(I - bu) \end{bmatrix} = \begin{bmatrix} A^+(I - bu) \\ u \end{bmatrix}\,.$$

Transposing the above equation yields to the promised equation. $\qquad\square$

### C.2  PROOF OF LEMMA 10

*Proof.* Let us first denote

$$v \triangleq (A^+)^\top x$$

and

$$G \triangleq (AA^\top)^{-1} \in \mathbb{R}^{n\times n}.$$

First note that by Cauchy-Schwarz inequality, it suffices to show there exists $\mathcal{D}$ such that $\mathbb{E}[\lambda_{\max}^4(G)] < +\infty$ and $\mathbb{E}\|v\|^4 < +\infty$.

We define $A_d \in \mathbb{R}^{n\times d}$ to be the submatrix of $A$ that consists of all $n$ rows and first $d$ columns. Denote

$$G_d \triangleq (A_d A_d^\top)^{-1} \in \mathbb{R}^{n\times n}.$$

We will prove $\mathbb{E}[\lambda_{\max}^4(G)] < +\infty$ by induction.

The base step is $d = n + 8$. Recall $\mathcal{D}_{[1:n+8]} = \mathcal{N}(0, I_{n+8})$. We first show $\mathbb{E}[\lambda_{\max}(G_{n+8})]^4 < +\infty$. Note that since $G_{n+8}$ is almost surely positive definite,

$$\mathbb{E}[\lambda_{\max}^4(G_{n+8})] = \mathbb{E}[\lambda_{\max}(G_{n+8}^4)] \leq \mathbb{E}\operatorname{tr}(G_{n+8}^4) = \mathbb{E}\operatorname{tr}((A_{n+8}A_{n+8}^\top)^{-4}) = \operatorname{tr}(\mathbb{E}[(A_{n+8}A_{n+8}^\top)^{-4}])\,.$$

By our choice of $\mathcal{D}_{[1:n+8]}$, the matrix $(A_{n+8} A_{n+8}^\top)^{-1}$ is an inverse Wishart matrix of size $n \times n$ with $(n+8)$ degrees of freedom, and thus has finite fourth moment (see, for example, Theorem 4.1 in (von Rosen, 1988)). It then follows that

$$\mathbb{E}[\lambda_{\max}^4(G_{n+8})] \leq \mathrm{tr}(\mathbb{E}[(A_{n+8} A_{n+8}^\top)^{-4}]) < +\infty \,.$$

For the inductive step, assume $\mathbb{E}[\lambda_{\max}(G_d)]^4 < +\infty$ for some $d \geq n+8$. We claim that

$$\lambda_{\max}(G_{d+1}) \leq \lambda_{\max}(G_d) \,,$$

or equivalently,

$$\lambda_{\min}(A_d A_d^\top) \leq \lambda_{\min}(A_{d+1} A_{d+1}^\top) \,.$$

Indeed, this follows from the fact that

$$A_d A_d^\top \preccurlyeq A_d A_d^\top + bb^\top = A_{d+1} A_{d+1}^\top \,,$$

under the Loewner order, where $b \in \mathbb{R}^{n \times 1}$ is the $(d+1)$-th column of $A$. Therefore, we have

$$\mathbb{E}[\lambda_{\max}^4(G_{d+1})] \leq \mathbb{E}[\lambda_{\max}^4(G_d)]$$

and by induction, we conclude that $\mathbb{E}[\lambda_{\max}^4(G)] < +\infty$ for all $d \geq n+8$.

Now we proceed to show $\mathbb{E}\|v\|^4 < +\infty$. We have

$$\|v\|^4 = \|(AA^\top)^{-1} Ax\|^4 \leq \|(AA^\top)^{-1} A\|_{op}^4 \cdot \|x\|^4 \,,$$

where $\|\cdot\|_{op}$ denotes the $\ell^2 \to \ell^2$ operator norm. Note that

$$\begin{aligned}
\|(AA^\top)^{-1} A\|_{op}^4 &= \lambda_{\max}^2 \left( \left( (AA^\top)^{-1} A \right)^\top (AA^\top)^{-1} A \right) \\
&= \lambda_{\max}^2 \left( A^\top (AA^\top)^{-2} A \right) \\
&= \lambda_{\max} \left( \left( A^\top (AA^\top)^{-2} A \right)^2 \right) \,,
\end{aligned}$$

where the last equality uses the fact that $A^\top (AA^\top)^{-2} A$ is positive semidefinite. Moreover, we deduce

$$\begin{aligned}
\|(AA^\top)^{-1} A\|_{op}^4 &= \lambda_{\max} \left( A^\top (AA^\top)^{-3} A \right) \\
&\leq \mathrm{tr} \left( A^\top (AA^\top)^{-3} A \right) \\
&= \mathrm{tr} \left( (AA^\top)^{-3} AA^\top \right) \\
&= \mathrm{tr} \left( (AA^\top)^{-2} \right) \,.
\end{aligned}$$

Using the fact that $A_d A_d^\top \preccurlyeq A_{d+1} A_{d+1}^\top$ established above, induction gives

$$(AA^\top)^{-2} \preccurlyeq (A_{n+8} A_{n+8}^\top)^{-2} \,.$$

It follows that

$$\mathbb{E}\left[ \|(AA^\top)^{-1} A\|_{op}^4 \right] \leq \mathbb{E}\left[ \mathrm{tr}\left( (A_{n+8} A_{n+8}^\top)^{-2} \right) \right] = \mathrm{tr}\left( \mathbb{E}\left[ (A_{n+8} A_{n+8}^\top)^{-2} \right] \right) < +\infty \,, \quad (13)$$

where again we use that fact that inverse Wishart matrix $(A_{n+8} A_{n+8}^\top)^{-1}$ has finite second moment.

Next, we demonstrate $\mathbb{E}\|x\|^4 < +\infty$. Recall that every $\mathcal{D}_i$ is either a Gaussian or a Gaussian mixture distribution. Therefore, every entry of $x$ has a subgaussian tail, and thus $\mathbb{E}\|x\|^4 < +\infty$. Together with (13) and the fact that $x$ and $A$ are independent, we conclude that

$$\mathbb{E}\|v\|^4 \leq \mathbb{E}\left[ \|(AA^\top)^{-1} A\|_{op}^4 \right] \cdot \mathbb{E}\left[ \|x\|^4 \right] < +\infty \,.$$

$\square$

### C.3 PROOF OF THEOREM 11

*Proof.* The randomness comes from $A, x, y$ and $b$. We first condition on $A$ and $x$ being fixed.

Let $G \triangleq (AA^\top)^{-1} \in \mathbb{R}^{n \times n}$ and $u \triangleq \frac{b^\top G}{1+b^\top Gb} \in \mathbb{R}^{1 \times n}$. Define

$$v \triangleq (A^+)^\top x, \quad r \triangleq 1 + b^\top Gb, \quad H \triangleq bb^\top.$$

We compute the left-hand side but take the expectation over only $y$ for the moment

$$\mathbb{E}_y \left\| \begin{bmatrix} A^\top \\ b^\top \end{bmatrix}^+ \begin{bmatrix} x \\ y \end{bmatrix} \right\|^2 - \|(A^+)^\top x\|^2$$

$$= \mathbb{E}_y \left\| (I - bu)^\top v + u^\top y \right\|^2 - \|v\|^2$$

$$= \|(I - bu)^\top v\|^2 + \mathbb{E}_y \|u^\top y\|^2 - \|v\|^2 \qquad (\mathbb{E}[y] = 0)$$

$$= \|(I - bu)^\top v\|^2 + \mathbb{E}_y[y^2] \frac{\|Gb\|^2}{r^2} - \|v\|^2.$$

Let us first consider the first and third terms of the above equation:

$$\|(I - bu)^\top v\|^2 - \|v\|^2 = v^\top \left( (I - bu)(I - bu)^\top - I \right) v$$

$$= -v^\top \left( bu + u^\top b^\top - buu^\top b^\top \right) v$$

$$= -v^\top \left( \frac{HG + GH}{r} - \frac{HG^2H}{r^2} \right) v.$$

Write $G = V\Lambda V^\top$, where $\Lambda = \mathrm{diag}(\lambda_1, \ldots, \lambda_n) \in \mathbb{R}^{n \times n}$ is a diagonal matrix ($\lambda_i > 0$) and $V \in \mathbb{R}^{n \times n}$ is an orthogonal matrix. Recall $b \sim \mathcal{N}(0, \sigma^2 I_n)$. Therefore $w \triangleq V^\top b \sim \mathcal{N}(0, \sigma^2 I_n)$. Taking the expectation over $b$, we have

$$\mathbb{E}_b \left[ \frac{HG + GH}{r} \right] = \mathbb{E}_b \left[ V \frac{V^\top bb^\top V\Lambda + \Lambda V^\top bb^\top V}{1 + b^\top V\Lambda V^\top b} V^\top \right] = V \mathbb{E}_w \left[ \frac{ww^\top \Lambda + \Lambda ww^\top}{1 + w^\top \Lambda w} \right] V^\top.$$

Let $R \triangleq \mathbb{E}_w \left[ \frac{ww^\top \Lambda + \Lambda ww^\top}{1 + w^\top \Lambda w} \right]$. We have

$$R_{ii} = \mathbb{E}_w \left[ \frac{2\lambda_i w_i^2}{1 + \sum_{i=1}^n \lambda_i w_i^2} \right] = \sigma^2 \mathbb{E}_{\nu \sim \mathcal{N}(0, I_n)} \left[ \frac{2\lambda_i \nu_i^2}{1 + \sigma^2 \sum_{i=1}^n \lambda_i \nu_i^2} \right] > 0$$

and if $i \neq j$,

$$R_{ij} = \mathbb{E}_w \left[ \frac{(\lambda_i + \lambda_j) w_i w_j}{1 + \sum_{i=1}^n \lambda_i w_i^2} \right].$$

Notice that for any $w$ and $j$, it has the same distribution if we replace $w_j$ by $-w_j$. As a result,

$$R_{ij} = \mathbb{E}_w \left[ \frac{(\lambda_i + \lambda_j) w_i (-w_j)}{1 + \sum_{i=1}^n \lambda_i w_i^2} \right] = -R_{ij}.$$

Thus the matrix $R$ is a diagonal matrix and

$$R = 2\sigma^2 \frac{\Lambda \, \mathrm{diag}(\nu)^2}{1 + \sigma^2 \nu^\top \Lambda \nu}.$$

Thus we get

$$\mathbb{E}_{b,A} \left[ \frac{HG + GH}{r} \right] = 2\sigma^2 \mathbb{E}_{\nu \sim \mathcal{N}(0, I_n), A} \left[ \frac{GV \, \mathrm{diag}(\nu)^2 V^\top}{1 + \sigma^2 \nu^\top \Lambda \nu} \right]$$

Moreover, by the monotone convergence theorem, we deduce

$$\lim_{\sigma \to 0^+} \mathbb{E}_{\nu \sim \mathcal{N}(0, I_n), A, x} \left[ -v^\top \frac{GV \, \mathrm{diag}(\nu)^2 V^\top}{1 + \sigma^2 \nu^\top \Lambda \nu} v \right] = \mathbb{E}_{\nu \sim \mathcal{N}(0, I_n), A, x} \left[ -v^\top GV \, \mathrm{diag}(\nu)^2 V^\top v \right]$$

$$= \mathbb{E}[-v^\top Gv].$$

It follows that as $\sigma \to 0^+$,

$$\mathbb{E}\left[-v^\top \frac{HG + GH}{r} v\right] \sim -2\sigma^2 \mathbb{E}[v^\top G v] = -2\sigma^2 \mathbb{E}\left[v^\top (AA^\top)^{-1} v\right] = -2\sigma^2 \mathbb{E}[\|(A^\top A)^+ x\|^2].$$

Moreover, by (4), we have

$$\mathbb{E}\left[v^\top (AA^\top)^{-1} v\right] \leq \mathbb{E}\left[\lambda_{\max}\left((AA^\top)^{-1}\right) \|(A^+)^\top x\|^2\right] < +\infty.$$

Next, we study the term $HG^2H/r^2$:

$$\mathbb{E}_{b,A}\left[\frac{HG^2H}{r^2}\right] = \mathbb{E}_{b,A}\left[V \frac{V^\top b b^\top V \Lambda^2 V^\top b b^\top V}{(1 + b^\top V \Lambda V^\top b)^2} V^\top\right]$$

$$= \mathbb{E}_{w \sim \mathcal{N}(0,\sigma^2 I_n),A}\left[V \frac{w w^\top \Lambda^2 w w^\top}{(1 + w^\top \Lambda w)^2} V^\top\right]$$

$$= \sigma^4 \mathbb{E}_{\nu \sim \mathcal{N}(0,I_n),A}\left[V \frac{\nu \nu^\top \Lambda^2 \nu \nu^\top}{(1 + \sigma^2 \nu^\top \Lambda \nu)^2} V^\top\right].$$

Again, by the monotone convergence theorem, we have

$$\lim_{\sigma \to 0^+} \mathbb{E}_{\nu \sim \mathcal{N}(0,I_n),A,x}\left[v^\top V \frac{\nu \nu^\top \Lambda^2 \nu \nu^\top}{(1 + \sigma^2 \nu^\top \Lambda \nu)^2} V^\top v\right]$$

$$= \mathbb{E}_{\nu \sim \mathcal{N}(0,I_n),A,x}\left[v^\top V \nu \nu^\top \Lambda^2 \nu \nu^\top V^\top v\right]$$

$$= \mathbb{E}_{A,x}\left[v^\top V \left(2\Lambda^2 + I_n \sum_{i=1}^n \lambda_i^2\right) V^\top v\right]$$

$$= \mathbb{E}\left[v^\top \left(2G^2 + \mathrm{tr}(G^2) I_n\right) v\right].$$

It follows that as $\sigma \to 0^+$,

$$\mathbb{E}_{b,A,x}\left[\frac{HG^2H}{r^2}\right]$$

$$\sim \sigma^4 \mathbb{E}\left[v^\top \left(2G^2 + \mathrm{tr}(G^2) I_n\right) v\right]$$

$$= \sigma^4 \mathbb{E}\left[2\|(AA^\top)^{-1} v\|^2 + \mathrm{tr}((AA^\top)^{-2}) \|v\|^2\right].$$

Moreover, by (4), we have

$$\mathbb{E}\left[2\|(AA^\top)^{-1} v\|^2 + \mathrm{tr}((AA^\top)^{-2}) \|v\|^2\right] \leq (n+2) \mathbb{E}\left[\lambda_{\max}^2((AA^\top)^{-1}) \|(A^+)^\top x\|^2\right] < +\infty.$$

We apply a similar method to the term $\frac{\|Gb\|^2}{r^2}$. We deduce

$$\frac{\|Gb\|^2}{r^2} = \frac{b^\top G^2 b}{(1 + b^\top Gb)^2} = \frac{b^\top V \Lambda^2 V^\top b}{(1 + b^\top V \Lambda V^\top b)^2}.$$

It follows that

$$\mathbb{E}\left[\frac{\|Gb\|^2}{r^2}\right] = \mathbb{E}_{w \sim \mathcal{N}(0,\sigma^2 I_n),A}\left[\frac{w^\top \Lambda^2 w}{(1 + w^\top \Lambda w)^2}\right] = \sigma^2 \mathbb{E}_{\nu \sim \mathcal{N}(0,I_n),A}\left[\frac{\nu^\top \Lambda^2 \nu}{(1 + \sigma^2 \nu^\top \Lambda \nu)^2}\right]$$

The monotone convergence theorem implies

$$\lim_{\sigma \to 0^+} \mathbb{E}_{\nu \sim \mathcal{N}(0,I_n),A}\left[\frac{\nu^\top \Lambda^2 \nu}{(1 + \sigma^2 \nu^\top \Lambda \nu)^2}\right] = \mathbb{E}[\nu^\top \Lambda^2 \nu] = \mathbb{E}[\mathrm{tr}(G^2)].$$

Thus we get as $\sigma \to 0^+$

$$\mathbb{E}_y[y^2] \frac{\|Gb\|^2}{r^2} \sim \sigma^4 \mathbb{E}[\mathrm{tr}(G^2)],$$

where $\mathbb{E}[\mathrm{tr}(G^2)] \leq n \mathbb{E}[\lambda_{\max}^2((AA^\top)^{-1})] < +\infty$.

Putting all three terms together, we have as $\sigma \to 0^+$

$$L_{d+1} - L_d \sim -2\sigma^2 \mathbb{E}[\|(A^\top A)^+ x\|^2].$$

Therefore, there exists $\sigma > 0$ such that $L_{d+1} - L_d < 0$. Furthermore, we deduce

$$L'_{d+1} - L'_d = \frac{1}{d^2}(L_{d+1} - L_d) < 0.$$

$\square$

## C.4 PROOF OF THEOREM 12

*Proof.* Again we first condition on $A$ and $x$ being fixed. Let $G \triangleq (AA^\top)^{-1} \in \mathbb{R}^{n \times n}$ and $u \triangleq \frac{b^\top G}{1+b^\top Gb} \in \mathbb{R}^{1 \times n}$ as defined in Lemma 9. We also define the following variables:

$$v \triangleq (A^+)^\top x, \quad r \triangleq 1 + b^\top Gb.$$

We compute $L'_{d+1} - L'_d$ but take the expectation over only $y$ for the moment

$$
\begin{aligned}
& \mathbb{E}_y \left\| \frac{1}{d+1} \begin{bmatrix} A^\top \\ b^\top \end{bmatrix}^+ \begin{bmatrix} x \\ y \end{bmatrix} \right\|^2 - \left\| \frac{1}{d}(A^+)^\top x \right\|^2 \\
&= \frac{1}{(d+1)^2} \left( \mathbb{E}_y \left\| (I-bu)^\top v + u^\top y \right\|^2 - (1+1/d)^2 \|v\|^2 \right) \\
&= \frac{1}{(d+1)^2} \left( \|(I-bu)^\top v\|^2 + \mathbb{E}_y \|u^\top y\|^2 - (1+1/d)^2 \|v\|^2 \right) && (\mathbb{E}[y] = 0) \\
&= \frac{1}{(d+1)^2} \left( \|(I-bu)^\top v\|^2 + \mathbb{E}_y[y^2] \frac{\|Gb\|^2}{r^2} - (1+1/d)^2 \|v\|^2 \right). && (14)
\end{aligned}
$$

Our strategy is to make $\mathbb{E}[y^2 \frac{\|Gb\|^2}{r^2}]$ arbitrarily large. To this end, by the independence of $y$ and $b$ we have

$$\mathbb{E}_{y,b}\left[ y^2 \frac{\|Gb\|^2}{r^2} \right] = \mathbb{E}_y[y^2] \mathbb{E}_b\left[ \frac{\|Gb\|^2}{r^2} \right].$$

By definition of $\mathcal{N}_{\sigma,\mu}^{\mathrm{mix}}$, with probability $2/3$, $y$ is sampled from either $\mathcal{N}(\mu, \sigma^2)$ or $\mathcal{N}(-\mu, \sigma^2)$, which implies $\mathbb{E}[y^2] \geq \frac{1}{3}\mu^2$. For each $b_i$, we have

$$\mathbb{P}(|b_i| \in [\sigma, 2\sigma]) \geq \frac{1}{3} \times \frac{1}{4}.$$

Also note that $G$ is positive definite. It follows that

$$\mathbb{E}_b\left[ \frac{\|Gb\|^2}{r^2} \right] = \mathbb{E}_b\left[ \frac{\|Gb\|^2}{(1+b^\top Gb)^2} \right] \geq \mathbb{E}_b \frac{(\lambda_{\min}(G)\|b\|)^2}{(1+\lambda_{\max}(G)\|b\|^2)^2} \geq \left( \frac{1}{12} \right)^n \frac{\lambda_{\min}^2(G) n\sigma^2}{(1+4\lambda_{\max}(G)n\sigma^2)^2}.$$

Altogether we have

$$\mathbb{E}_{y,b}\left[ y^2 \frac{\|Gb\|^2}{r^2} \right] \geq \frac{1}{3 \cdot 12^n} \frac{n\lambda_{\min}^2(G)\mu^2\sigma^2}{(1+4n\lambda_{\max}(G)\sigma^2)^2}.$$

Let $\mu = 1/\sigma^2$ and we have

$$
\begin{aligned}
\lim_{\sigma \to 0^+} \mathbb{E}\left[ y^2 \frac{\|Gb\|^2}{r^2} \right] &\geq \lim_{\sigma \to 0^+} \mathbb{E}_{A,x} \mathbb{E}_{y,b}\left[ \frac{1}{3 \cdot 12^n} \frac{n\lambda_{\min}^2(G)}{\sigma^2(1+4n\lambda_{\max}(G)\sigma^2)^2} \right] \\
&= \mathbb{E}_{A,x}\mathbb{E}_{y,b} \lim_{\sigma \to 0^+}\left[ \frac{1}{3 \cdot 12^n} \frac{n\lambda_{\min}^2(G)}{\sigma^2(1+4n\lambda_{\max}(G)\sigma^2)^2} \right] \\
&= +\infty,
\end{aligned}
$$

where we switch the order of expectation and limit using the monotone convergence theorem. Taking full expectation over $A, x, b$ and $y$ of (14) and using the assumption that $\mathbb{E}\|v\|^2 < +\infty$ we have

$$L'_{d+1} - L'_d = \frac{1}{(d+1)^2}\left( \mathbb{E}_{A,x,b}\|(I-bu)^\top v\|^2 + \mathbb{E}\left[ y^2 \frac{\|Gb\|^2}{r^2} \right] - (1+1/d)^2 \mathbb{E}_{A,x}\|v\|^2 \right) \to +\infty$$

as $\sigma \to 0^+$. In addition, we have as $\sigma \to 0^+$,

$$L_{d+1} - L_d \geq d^2(L'_{d+1} - L'_d) \to +\infty.$$

$\square$

