# OpenReview forum: "Multiple Descent: Design Your Own Generalization Curve"
_ICLR.cc/2021/Conference — Reject_

### Official Review · AnonReviewer2 · 2020-10-25
**My decision hinges on a very convincing explanation on the model settings and motivation. Currently I consider a reject.**

**Rating:** 5
**Confidence:** 4

**Review:**

This paper studies the double/multiple descents of the prediction error curve for linear regression in both the under-parameterized and over-parameterized regime.

The strength: while many papers have studied the double descent for linear regression estimate or the minimum $\ell_2$ norm solution,  this paper shows multiple descents when d=O($\sqrt{n}$) a setting is barely studied by others. Further, while multiple descents have been numerically discovered by other concurrent works, they have theoretically proved that such multiple descents exist.

The weakness: The major weakness of the paper is the model settings. Specifically, 1) it is unclear why the prediction error is normalized by the number of features, and 2) the bias term is left out in the prediction error due to the true coefficients being zero and only the variance term is considered. First, for normalization, the authors claim that this normalization is necessary for comparison. Indeed, the entire results are hinged on this normalization, i.e., without the normalization, the proof can NOT show the existence of the multiple descents neither in under-parameterized regime nor overparameterized regime. The reasons I found this normalization is weird are the following:

i) Normal linear regression problem does not have such normalization on the prediction error. It is unclear why we want to divide a one-dimensional error by the feature size.

ii) Other double descent works mainly deliver two messages:

a) Given a fixed sample size, what is the best model gives the best estimate of the response. The answer is a larger model, i,e, adding more features, may help. (e.g. Xu's PCR paper)

b) Given a fixed feature size, what is the best sample size that gives the best estimate of the response. The answer is using a smaller sample size may help. (e.g. Hastie's double descent paper)

For both cases, I do not see any reason to normalize the prediction error of response by feature size. If this normalization is for the purpose of the model selection penalty, it is unclear why we should encourage a larger model instead of penalizing it.

A reasonable quantity for such normalization is the MSE of the coefficient, i.e., $\|\hat{\beta}-\beta^*\|^2$. There are many applications where people are more interested in the coefficients rather than the response. Maybe the authors should consider this quantity instead of the prediction error.

For the second weakness of the model settings, the bias term has been left out of the prediction error when the true coefficients are assumed to be all zero. Because of this setting, all features are just pure noise, irrelevant to the response. Then, we can check that 0 is the best estimate when all features are just pure noise, and it seems that there is no motivation for us to learn anything from the random noise. If the main purpose of this paper is to deliver a message that using only irrelevant features and adding more of them can help to improve the prediction error, this effect is known already in those double descent paper in the overparameterized regime. Showing multiple descents does not add much value because it never beats the trivial estimate 0 in this setting.

Because of these major weakness, I recommend rejection for this paper. But I will possibly change my evaluation if the authors can provide a very convincing explanation of the model settings and motivation.

Besides these, another suggestion for the paper is that the proof of the Theorems and the statement of Lemmas takes a lot of places. I think they can be replaced by more detailed discussions of the model settings and messages or conclusions from the main theorems. For example, is there any intuition about what kind of multiple descents curve is more favorable? Also, despite the attractive title, I think it is still hard to design the generalization curve without taking the bias term into consideration. The room can be left for the analysis of the bias term.

After response:
Thanks for addressing the concern about normalization. It appears that other reviewers have a concern about such normalization as well. I suggest the authors remove the results with normalization entirely from the main paper and only have it in the appendix for anyone that is interested in such normalization.

On the other hand, without normalization, the results have changed for the under-parameterized regime (which makes more sense to me) and the proof looks quite different in the over-parameterized regime as well. I did not have time to check the proof and I believe it is better to resubmit the paper as new because of the major changes.

Finally, I still have concerns about the fact that only variance is discussed. I suggest the authors state their results in a setting where both bias and variance exists and the features added to the model are related to the response. Otherwise, it is a weird message that it is good to add pure noise as features. It feels like although we can design multiple descents in the overparameterized regime when noise is large, it is very likely that the 0 estimate achieves the best prediction risk. So there is no point to go into overparameterization and multiple descents at all.

In summary, I have raised the score to 5. I believe it can be 6 or 7 if all issues are addressed, but I am afraid that the paper looks basically new after these changes and thus I am not sure whether it should be still considered for this conference.

---

> ### Author Response · Authors · 2020-11-23
> **Reesponse to Reviewer 2**
>
> Q: The prediction error should not be normalized. The entire results are hinged on this normalization.
>
> A: This is a common and main concern of reviewer 2,3 and 4. We fully appreciate the reviewers’ comment and thank them for pointing it out. We would like to remark that our proof technique is general enough that we obtain similar multiple-descent results for the un-normalized generalization error. Therefore, we have uploaded an updated version giving proofs for the un-normalized case.
>
> 1. For the underparameterized regime, our updated proof shows that the generalization error always increases and this is regardless of the data distribution (please see Thm 1) and also regardless of whether true $\beta$ is zero or not (please see Remark 1). This result matches the corresponding result in the double descent paper by Belkin et al., as mentioned by the reviewer. Please note that this is a different result from the old version.
> 2. For the overparameterized regime, the generalization curve still has multiple descent, whether the error is normalized or not. Please see Thm 8.
>
> Q: The bias term is left out in the prediction error due to the true coefficients being zero and only the variance term is considered.
>
> A: This assumption that $\beta$ is zero is only there to simplify the analysis. Our results can be generalized to the case where the true parameter is not zero. Currently we have added a remark (Remark 1) in the problem setup of the updated version explaining why the results generalize. In the future version we will give a more detailed discussion of the general case where $\beta$ is nonzero. In the following we will briefly explain why whether $\beta$ is zero or not does not affect the results.
>
> 1. In the under-parameterized regime, the updated proof has already shown that including the bias term will not hurt our result, as is pointed out in the answer to the question regarding normalization.
>
> 2. In the overparameterized regime where the loss is un-normalized, the generalization curve still shows the multiple-descent behavior, which is essentially the same as the current results. This is because the loss consists of a variance term which is what we consider in the paper and a bias term which is currently omitted by assuming that $\beta$ is zero. The current proof shows the variance term has multiple descent. Note that in the overparameterized case the bias term always decreases as the dimension increases if Gaussian entries are added. Therefore, to make the loss (i.e., variance + bias) decrease, no extra effort is needed. But how to make the loss increases? It is also easy: as is shown by Thm 12, the increase in the variance term can be arbitrarily large, and thus can be made large enough to offset the decrease in the bias.
>
> To conclude, our results hold true in the general case where $\beta$ is nonzero and the loss is un-normalized.
>
> Q: The proof of the Theorems and the statement of Lemmas takes a lot of places. They can be replaced by more detailed discussions of the model settings and messages or conclusions from the main theorems.
>
> A: Great suggestion. We have added more discussion to the updated version and we will add more discussion in the future versions of the paper.

---

### Official Review · AnonReviewer4 · 2020-10-27
**The paper studies an interesting problem but gives an unsatisfactory answer to it.**

**Rating:** 4
**Confidence:** 4

**Review:**

Short summary:
The paper claims that the double descent phenomenon arises from 'specific interactions between properties of typical data and the inductive biases of algorithms' and constructs a distribution that generates an arbitrary generalization curve for linear regression, specifically building a multiple descent generalization curve both in the under and overparametrized regimes.

The model that is used in the paper is a linear regression model over an increasing (in a revealing manner) set of coordinates or features. The authors construct the distribution that gives the peaks at custom coordinates by having features being independent  standard normal when they want the test error to decrease and to be a (independent) mixture of gaussians when they want the test to increase.
++++++
Main points:
While the math to my understanding is clean and the exposition is clear, my main concern is how the authors relate their findings to Double Descent. This worries me in two related ways. First, from the perspective of the complexity of the model.  While adding a dimension to the linear regression adds a parameter, I'm skeptical how this relates to the complexity of the model in how we view complexity in machine learning and in the research area of double descent in particular. I would be much more convinced if the authors could show a case where adding a feature in the random features sense where the features are of the whole vector (say apply a random rotation and then do the inverse transform sampling) or adding a neuron in a two layer network and still being able to decrease/increase performance arbitrarily (or close to it in some sense).  Even doing the same as in https://arxiv.org/pdf/1903.07571.pdf, where they choose a random set of indices of increasing cardinality would convince me much more. The second related issue, is the distribution of the features. I would not mind it if the classifier would use the features uniformly, but increasing/decreasing the hardness of the distribution at each coordinate feels very artificial in the following sense: Assume that the first coordinate is the label (or something close to it), but the next coordinates are pure noise. Then both our train and test will increase when we increase the number of features. In my intuition, this is very far from what is studied and claimed in the double descent literature (for example in the sense of Belkin's interpolation point or Nakkiran's Model Complexity, we expect the train error to decrease when model capacity increases).

I do believe that the question of whether we can construct an arbitrary generalization curve is very important and that it should be studied and explored more deeply, but I'm not convinced by the set-up in this paper. I would be willing to change my opinion in the case the authors will address the above points in a satisfactory manner.

Minor comments:
1) The related work in the body of the paper is lacking:
  (i)  One notable paper that should be present is: Advani & Saxe 17' https://arxiv.org/abs/1710.03667.
  (ii) While Nayshabur 15' observe the double decent without realizing it and Neal 18' study the bias-variance tradeoff, Nakkiran 19' https://arxiv.org/abs/1912.02292 is the first to demonstrate it in a convincing fashion and should be cited as such.
2) I would appreciate an explanation for why the loss is scaled by $1/d^2$, this feels rather arbitrary.

---

> ### Author Response · Authors · 2020-11-23
> **Response to Reviewer 4**
>
> Q: how adding a dimension in the linear regression relates to the complexity of the model in how we view complexity in machine learning and in the research area of double descent in particular?
>
> A: In the linear regression model, adding one dimension to the features is the same as increasing the dimension of the model parameter, which gives the model more power to fit to the given data. Therefore, to some extent, we think the feature dimension can be viewed as a surrogate for the model complexity, just like many other more commonly used surrogates like Rademacher complexity. And we totally agree that the data dimension is definitely not equivalent to the model complexity.
>
> Q: Regarding the distribution of the features: increasing/decreasing the hardness of the distribution at each coordinate feels very artificial.
>
> A: The data distribution for each dimension is fixed in our paper. Each data point is then generated in an iid fashion, where the entries follow the corresponding pre-specified feature distributions (this is the standard learning model). Here, the feature distribution is either Gaussian or Gaussian mixture. As we increase the dimension of the parameter, more features are revealed. One should not confuse the data distribution with the feature revealing process.
>
> Q: The prediction error should not be normalized.
>
> A: This is a common and main concern of reviewer 2,3 and 4. We fully appreciate the reviewers’ comment and thank them for their acute observation.  In order to compare quantities of different dimensions, we decided to normalize the generalization loss by the dimension.  However, we would like to remark that our proof technique is general enough that we obtain similar multiple-descent results for the un-normalized generalization error as well. Therefore, we have uploaded an updated version giving proofs for the un-normalized case.
>
> 1. For the underparameterized regime, our updated proof shows that the generalization error always increases and this is regardless of the data distribution (please see Thm 1) and also regardless of whether true $\beta$ is zero or not (please see Remark 1). This result matches the corresponding result in the double descent paper by Belkin et al., as mentioned by the reviewer. Please note that this is a different result from the old version.
> 2. For the overparameterized regime, the generalization curve still has multiple descent, whether the error is normalized or not. Please see Thm 8.
>
> Q: Some related works need to be added.
>
> A: We have added the related works mentioned. Thanks for pointing out.

---

> > ### Comment · AnonReviewer4 · 2020-11-25
> > **Thanks for the response.**
> >
> > * I would like to thank the Authors for their response. I agree with the authors that the number of features can be viewed as a surrogate for model complexity and glad that we are in agreement to whether that data dimension is not equivalent to the model complexity.
> >
> > * The issue in my opinion is not whether the distribution is fixed or not. The issue that when combining the set up with the distribution, the problem (distribution) becomes harder. Hence the axis that is usually reserved for model complexity in the double descent plot, becomes hardness of distribution or at least it is a major confounding factor that has to be addressed. This is because when the distribution becomes harder (or easier) it is clear that generalization error will decrease (or increase).
> >
> > Thus, my main concern (and as I mentioned before, my assessment of the paper hinges on that) is that the set up is not convincing with respect to monotonicity of model's complexity as opposed to conflating factors such as distribution difficulty.
> >
> > * The normalization just seemed arbitrary to me, not a major concern--thank you for addressing it nevertheless. Also, I appreciate the addition of the related work.

---

### Official Review · AnonReviewer1 · 2020-10-28
**Important result; mismatched settings;**

**Rating:** 6
**Confidence:** 3

**Review:**

Previous work has shown peaks in generalization error as the capacity of the model increases (called the double-descent phenomenon). The submitted paper proposes methods for generating data that would arbitrarily change the number and positions of peaks in a generalization-vs-capacity curve for linear regression, where the number of features controls the capacity,  and shows that properties of data can play an important role in this phenomenon.

This paper tackles an important problem in a quite active area of research with clear presentation and coherent organization. Existence of this data serves as an impossibility result that shows that relating the double descent phenomenon to the properties of model and interpolation without further assumptions on the data is futile. However, there is a critical discrepancy between the generalization curves studied in this paper and previous work that I describe below and, therefore, I'm leaning towards rejection. I will raise my score if the authors can show that the effects on the number and positions of the peaks hold in the original setting, as I believe this is an important paper otherwise.

The generalization error in this paper is normalized by the square of the number of features and this can have major effects on the shape of generalization-vs-capacity curve. The number of features is what controls the capacity so, for example, if the regular (unnormalized) error is flat across different capacities, the normalized curve will be a decreasing sequence.

Neither the generalization error in a classical bias-variance curve nor the error that matters to a practitioner is normalized. I skimmed through the double descent paper by Belkin et al and they also seem to be using the typical generalization error which is not normalized.

The motivation for normalization in the paper is that the closed form error, $||(A^\top)^+x||^2$, sums over d dimensions and so the generalization error has to be normalized by d^2. This does not seem right. $(A^\top)^+$ itself has factors that sum over d dimensions and are then inverted, so the effect of d will cancel out.

Minor remarks:
- \beta and A are not clearly defined in the problem setup.

--
Update: The issue with normalization is fixed in the new version and I am increasing my score.

---

> ### Author Response · Authors · 2020-11-23
> **Response to Reviewer 1**
>
> Q: Error should not be normalized.
>
> A: This is a common and main concern of reviewer 2,3 and 4. We fully appreciate the reviewers’ comment and thank them for their acute observation. We would like to remark that our proof technique is general enough that we obtain similar multiple-descent results for the un-normalized generalization error. Therefore, we have uploaded an updated version giving proofs for the un-normalized case.
>
> To be more specific:
>
> 1. For the underparameterized regime, the generalization error always increases and this is regardless of the data distribution (please see Thm 1) and also regardless of whether true $\beta$ is zero or not (please see Remark 1). This result matches the corresponding result in the double descent paper by Belkin et al., as mentioned by the reviewer. Please note that this is a different result from the old version.
> 2. For the overparameterized regime, the generalization curve still has multiple descent, whether the error is normalized or not. Please see Thm 8.

---

### Official Review · AnonReviewer3 · 2020-10-28
**Generalization loss design for linear regression**

**Rating:** 6
**Confidence:** 2

**Review:**

##########################################################################
Summary:

The paper proves that the generalization curve of a linear regression problem can be designed. The paper discusses both the under-parameterized and over-parameterized case and shows that the generalization curve can be designed in either case. The paper presents only theoretical results.


##########################################################################
Reason for score:

My vote is for accepting the paper. The subject it addresses is of importance and I believe the results that are presented are of sufficient interest.

##########################################################################
Pros:

1. The generalization error is an important aspect for ML algorithms. The paper addresses the case of linear regression, one of the simplest ML algorithms. However, showing that the generalization error can be controlled even for a simple model as this is nonetheless important.

2. The paper is well written, the problem it addresses is clearly discussed and the development of the proposed method is well detailed.


##########################################################################
Cons:

1. I would have liked to have some numerical examples to illustrate the design of the generalization curve for a simple case.

2. In the setting in the paper you draw the new elements either from a normal distribution or from a mixture distribution when you increase the dimension. In a practical settings, where I already have the data, do such hypothesis still remain true?


##########################################################################
Miscellaneous:

1. Could you please elaborate on the statement that 'the true linear model is \beta = 0 \in R^{d}'. For me it is not clear what is the purpose of the statement, do you mean that the model parameters are all zero?

2. There are some typos present, for example 'The quantiry ...' in the paragraph after lemma 3, they should all be spotted by a spell checker.

---

> ### Author Response · Authors · 2020-11-23
> **Response to Reviewer 3**
>
> Q: In the setting in the paper you draw the new elements either from a normal distribution or from a mixture distribution when you increase the dimension. In a practical settings, where I already have the data, do such hypothesis still remain true?
>
> A: Gaussian and Gaussian mixture distributions are very common assumptions and also widely considered in the machine learning and statistical literature. Moreover, we would also like to remark that the results hold for more general distributions than just Gaussian and Gaussian mixture distributions.
>
> Q:  Could you please elaborate on the statement that 'the true linear model is $\beta = 0 \in \mathbb{R}^{d}$'. For me it is not clear what is the purpose of the statement, do you mean that the model parameters are all zero?
>
> A: Yes, by $\beta = 0\in\mathbb{R}^{d}$ we are assuming that the true model parameters are all zero. However, this assumption is only to simplify the analysis. Our results can be generalized to the case where the true parameter is not zero.

---

### Author Response · Authors · 2020-11-21
**General comment**

Dear Reviewers and AC,

We sincerely appreciate all the reviews. We also agree that studying the multiple descent phenomenon is an important problem. We are working on incorporating the insightful and valuable suggestions from the reviewers. We will update the draft and post the individual responses very soon.

Thanks.

---

### Author Response · Authors · 2020-11-23
**Revised Version**

In the revised version of the manuscript, we have incorporated the reviewers’ comments. In particular, we added the proof of multiple descent for the unnormalized generalization error.  In the revision, we use $L_d$ to denote the unnormalized generalization error and $L^\prime_d$ the normalized one.

In Theorem 1 we prove that in the underparameterized regime, the unnormalized generalization error $L_d$ always increases with dimension $d$ and that the gap $L_{d+1}-L_d$ can be made arbitrarily large.

In Theorem 8, we prove that in the overparameterized regime, one can control both the unnormalized generalization error $L_d$ and the normalized one $L^\prime_d$ going up and down in the desired dimensions. The proof of Theorem 8 uses our revised Theorems 11 and 12, in which we prove that we can make the gaps $L_{d+1}-L_d$ and $L’_{d+1}-L_d$ negative or arbitrarily large.

---

### Decision · Program_Chairs · 2021-01-07
**Final Decision**

**Decision:**

Reject

**Comment:**

While there was some interest in the analysis, the consensus view was that the original treatment was not sufficiently well-motivated, and the revision was too dissimilar from the original submission for it to be evaluated for publication in this year's ICLR.